# Gap junctions mediate discrete regulatory steps during fly spermatogenesis

**Yanina-Yasmin Pesch**[1], **Vivien Dang**[1], **Michael John Fairchild**[1¤], **Fayeza Islam**[1], **Darius Camp**[1], **Priya Kaur**[1], **Christopher M. Smendziuk**[1], **Anat Messenberg**[1], **Rosalyn Carr**[1,2], **Ciaran R. McFarlane**[3], **Pierre-Yves Musso**[1], **Filip Van Petegem**[3], **Guy Tanentzapf**[1]*

1 Department of Cellular and Physiological Sciences, University of British Columbia, Vancouver, Canada, 2 School of Biomedical Engineering, University of British Columbia, Vancouver, Canada, 3 Department of Biochemistry, University of British Columbia, Vancouver, Canada

¤ Current address: Stem Cell Program and Division of Hematology/Oncology, Boston Children's Hospital and Dana-Farber Cancer Institute, Howard Hughes Medical Institute, Harvard Stem Cell Institute, Harvard Medical School, Boston, Massachusetts, United States of America

* tanentz@mail.ubc.ca

**Data Availability Statement:** All relevant data are within the manuscript and its Supporting Information files.

**Funding:** Funding for this study was provided by a grant to GT from the Natural Sciences and

## Abstract

Gametogenesis requires coordinated signaling between germ cells and somatic cells. We previously showed that Gap junction (GJ)-mediated soma-germline communication is essential for fly spermatogenesis. Specifically, the GJ protein Innexin4/Zero population growth (Zpg) is necessary for somatic and germline stem cell maintenance and differentiation. It remains unknown how GJ-mediated signals regulate spermatogenesis or whether the function of these signals is restricted to the earliest stages of spermatogenesis. Here we carried out comprehensive structure/function analysis of Zpg using insights obtained from the protein structure of innexins to design mutations aimed at selectively perturbing different regulatory regions as well as the channel pore of Zpg. We identify the roles of various regulatory sites in Zpg in the assembly and maintenance of GJs at the plasma membrane. Moreover, mutations designed to selectively disrupt, based on size and charge, the passage of cargos through the Zpg channel pore, blocked different stages of spermatogenesis. Mutations were identified that progressed through early germline and soma development, but exhibited defects in entry to meiosis or sperm individualisation, resulting in reduced fertility or sterility. Our work shows that specific signals that pass through GJs regulate the transition between different stages of gametogenesis.

## Author summary

Gap-junctions allow neighboring cells to communicate by connecting their cytoplasm. Gap-junctions play an essential role during sperm development by facilitating communication between the two cell types found in the testes, the germline which produces sperm, and the soma, which provides an essential supportive environment to the germline. We sought to better understand the ways in which gap-junctions help germline and somatic cells to communicate. We introduced nearly twenty different mutations into a gap-

Engineering Research Council of Canada (NSERC, RGPIN-2018-04648; https://www.nserc-crsng.gc.ca/index_eng.asp) and to FVP from the Canadian Institute for Health Research (CIHR, CIHR PJT-159601; https://cihr-irsc.gc.ca/e/193.html). The funders had no role in study design, data collection and analysis, decision to publish, or preparation of the manuscript.

**Competing interests:** The authors have declared that no competing interests exist.

junction gene that connects the soma and germline in the fly testes. These mutations were chosen based on bioinformatics and analysis of the predicted structure of the gap-junction protein. We replaced the normal version of the gap-junction with the mutated versions in flies, and analysed how sperm development was affected. Based on this analysis we identified key parts of the protein that were required for the assembly and maintenance of the gap-junctions. Moreover, mutations designed to selectively disrupt the passage of specific materials through the gap-junction blocked different stages of sperm development. Mutations were identified that progressed through early sperm development, but exhibited defects in later stages, resulting in sterility. Our work shows that specific signals that pass-through gap-junctions regulate the transition between different stages of sperm development.

## Introduction

In animals two tissue types populate the gonads, the germline, which gives rise to the gametes, and the soma, which gives rise to all other tissues that support and maintain gamete formation. Gametogenesis is a complex process that requires the intricate cooperation of the soma and germline. The soma supports and nourishes the germline [1–4], provides signals for stem cell niche formation and gamete differentiation [2,5–7], and forms the architectural framework for gametogenesis [8]. Successful gametogenesis requires ongoing communication between soma and germline and when this communication is disrupted this results in infertility or tumor formation [2,9,10].

Spermatogenesis in *Drosophila* has proven to be a versatile, genetically tractable, model system for studying soma and germline communication [11–13]. The *Drosophila* testis is a blind-ended coiled tube that contains a stem cell niche, called the hub, at its tip [7,14]. The hub, composed of a cluster of 8–15 somatic cells, has two main functions: first, it physically anchors both germline stem cells (GSCs) and somatic cyst stem cells (CySCs) and second, it secretes molecules that regulate and keep GSCs and CySCs in an undifferentiated state [4,7,14]. As the GSCs divide asymmetrically [15], the daughter cell, called a gonialblast, is displaced from the hub, which enables it to differentiate [7,11] and undergo mitotic transit amplifying divisions to syncytial spermatogonia [12]. Once clusters of 16 interconnected spermatocytes are formed, they enter meiotic divisions and initiate a differentiation program, resulting in 64 connected spermatids [16]. Spermatids undergo dramatic morphological changes, including elongation and individualization, to form mature sperm, which is then stored in the seminal vesicle [16]. CySCs also divide asymmetrically, giving rise to cyst cells, two of which surround and encapsulate each gonialblast [9,10,17,18]. Through encapsulation, the developing germ cells are fully surrounded by somatic cells, completely isolating them from outside cues [9,19]. This makes cell communication between soma and germline indispensable for the delivery of regulatory signals and nutrients to the developing germline.

Gap junctions (GJs) are transmembrane channels encoded by Innexins in invertebrates and Connexins in vertebrates [20,21], these two protein families share significant structural homology, but limited sequence homology [22]. The true vertebrate homologue of Innexins are not Connexins but rather Pannexins. These are channel forming proteins that do not form GJs but rather functional hemichannels, and share significant sequence homology with Innexins [23]. While hemichannels do not connect adjacent cells, GJs form when two hemichannels on neighboring cells link up to form an active channel allowing the passage of directly from the one cell to another. Although it is known that Innexins form GJ, it is not known whether, like

Pannexins, their true homologs, they also form functional hemichannels. The linkage between Connexin or Innexin in neighboring cells, called docking, occurs via disulfide bridges between cysteine residues located in the extracellular part of the GJ proteins [22,24]. GJs allow the passage of molecules smaller than 1 kDa, for Connexins, and 3 kDa, for Innexins [25], and known cargos include ions ($Ca^{2+}$) and second messengers ($IP_3$, cAMP) [26]. The passage of cargo through the channel is highly controlled and can be regulated by the opening and closing of the channel, a function referred to as gating. Gating of Connexins is modulated by changes in pH, calcium concentration [27–29], and voltage within the channel pore [30]. Connexins, Innexins, and Pannexins are 4 pass transmembrane proteins with intracellular C- and N-termini domains [20,22,31]. While the C-terminal intracellular domain of connexins is known to regulate channel gating, it is also known to have channel independent functions. Specifically, the C-terminus is an important docking point for cytoplasmic proteins and is also subjected to post-translational modifications such as phosphorylation, that influence intracellular trafficking and signaling [32–34]. The N-terminal intracellular domain of connexins may also play a role in channel gating [28,35]. For example, the N-terminus of connexins has been shown to influence channel conductance, permeability, and voltage-dependent gating [36]. Also, the N-terminal intracellular domain of Connexin 26 has been shown to undergo a pH dependent conformational change that controls gating [37]. Mutations in the N-terminal domain of Connexin have been implicated in multiple human diseases, suggesting it plays a key role in dysregulation of connexins and the etiology of gap junction-associated diseases. Human pathologies associated with mutations in the N-terminal domain of connexin include KID (Keratitis-Ichthyosis-Deafness) syndrome [38,39], X-linked Charcot-Marie-Tooth disease [40] and hereditary eye cataracts [41].

Gap junctions are involved in soma-germline communication in many organisms. In *C. elegans*, different innexin proteins localize to the soma-germline interface and are required for proliferation and differentiation of GSCs as well as regulation of oocyte maturation [42]. In mammalian testes, gap junction-mediated soma-germline communication was shown to play a crucial role for spermatogenesis and fertility [43]. Connexins can be found connecting different cell types in the testis, notably the developing germ cells and somatic Sertoli cells [44], as well as Sertoli cells and hormone producing Leydig cells [45]. The transport of cargo is thought to occur unidirectionally from somatic Sertoli cells to developing spermatogonia and spermatocytes [46]. Loss of Connexin43 (Cx43) from murine Sertoli cells leads to hyperplasia of Leydig cells indicating crosstalk between the two cell types [45] and subsequently, to arrested germ cell differentiation at the spermatogonia stage [47,48]. A transcriptomic analysis in human patients suffering from Sertoli Cell Only (SCO) Syndrome, a severe form of infertility in men characterized by complete absence of germ cells, showed strongly reduced expression of Cx26, which in mice regulates crosstalk between Sertoli cells and spermatogonia [49,50]. These examples from different species indicate that soma-germline communication through gap junctions is a conserved mechanism.

The *Drosophila* the gap junction protein Zpg (Zero population growth, Inx4) localizes to the plasma membrane of germ cells and is required for fertility. Male and female flies lacking Zpg have rudimentary testes and ovaries, respectively, and are sterile [51]. In female flies, Zpg is required in germ cells for their maintenance as well as for the early stages of their differentiation [51,52]. In the testes, Zpg couples to Inx2 in neighboring somatic cells and forms a channel composed of two different innexins, known as a heterotypic channel, that is required for germ cell maintenance and for regulating proliferation and differentiation of both germ and somatic cells [53]. However, the precise mechanism of action of these gap junctions in soma-germline communication is not well understood and the nature of the signal that is being transmitted through the gap junctions is not known.

To elucidate how Zpg regulates stem cell maintenance and differentiation in the fly testes, we carried out systematic structure-function analysis of Zpg using information derived from bioinformatics and structural biology approaches. To this end, we replaced endogenous Zpg with a collection of mutant versions of the protein, affecting key domains and residues, including those predicted to control membrane trafficking, C-terminal phosphorylation, coupling to Inx2, and channel gating. Our results establish a mechanistic framework for Zpg activity in the testes, by identifying residues that are indispensable for its trafficking to the membrane and its coupling to other innexins to produce functional gap junctions. Importantly, a set of point mutations that were introduced to modulate channel-gating gave rise to unique phenotypes that act at discrete steps in the developmental sequence of sperm production. This shows that specific gap-junction mediated signals control the stepwise progression of germ cell differentiation during spermatogenesis.

## Results

### A genomic rescue construct for zpg allows a detailed structure/function analysis

In order to carry out a detailed structure-function analysis of the Zpg protein we relied on a previously identified genomic fragment of approximately 6kb that was shown to be sufficient for complete rescue of the *zpg* mutant phenotype [51, 53]. Close analysis of the genomic region of the *zpg* gene (Fig 1A) showed that the *zpg* rescue construct contains the complete coding sequence of Zpg as well as the annotated 3' and 5' UTR regions. Further support for the idea that the rescue construct contains the entire coding and regulatory regions required for *zpg* function comes from the location of the *zpg* gene within an intron for the gene *rexo5* and the ability of the rescue construct to compensate for the loss of the endogenous *zpg* gene [53] (see below). We previously showed that tagging the rescue construct by the addition of a GFP to the C-terminal domain had no impact on the ability of the construct to rescue *zpg* null mutants (see Materials and Methods; [53]). Flies containing this GFP-tagged genomic rescue construct were introduced into the *zpg* null mutant background, giving rise to a viable line which we refer to as *zpg*::GFP GR (GR for genomic rescue).

### Rationale for selecting residues to target in a structure/function analysis based on sequence conservation and protein structure

Multiple factors were used to identify candidate residues for targeting in a structure function approach. Specifically, information was derived from three independent sources: sequence alignments, analysis of a homology-modelling derived protein structure of Zpg, and previous biochemical and mutational studies of innexins and/or connexins. Gap junction proteins exhibit similarities in their internal domain arrangement and as well as their structural homology [20, 22]. Sequence alignments among the 8 *Drosophila* innexin proteins as well as between *Drosophila* and *C. elegans* innexins were used to identify residues of interest (see for example an alignment of the N-terminal domains on fly and worm innexins in Fig 1B). The level of sequence identity between *Drosophila* Zpg and *C. elegans* INX-6, (~30%), was sufficient to allow homology modelling, a methodology that has been successfully used before for structure-function studies of gap junction proteins [54]. The *C. elegans* INX-6 was used for homology modelling as it is currently the only known CryoEM structure of an Innexin (see materials and methods; [55]). The Cryo-EM structure of *C. elegans* INX-6 [55, 56] provides intriguing clues about how the passage of cargoes through innexins is regulated. For example, the INX-6 structure showed that the N-terminal region as well as the Extracellular Helix-1 region face

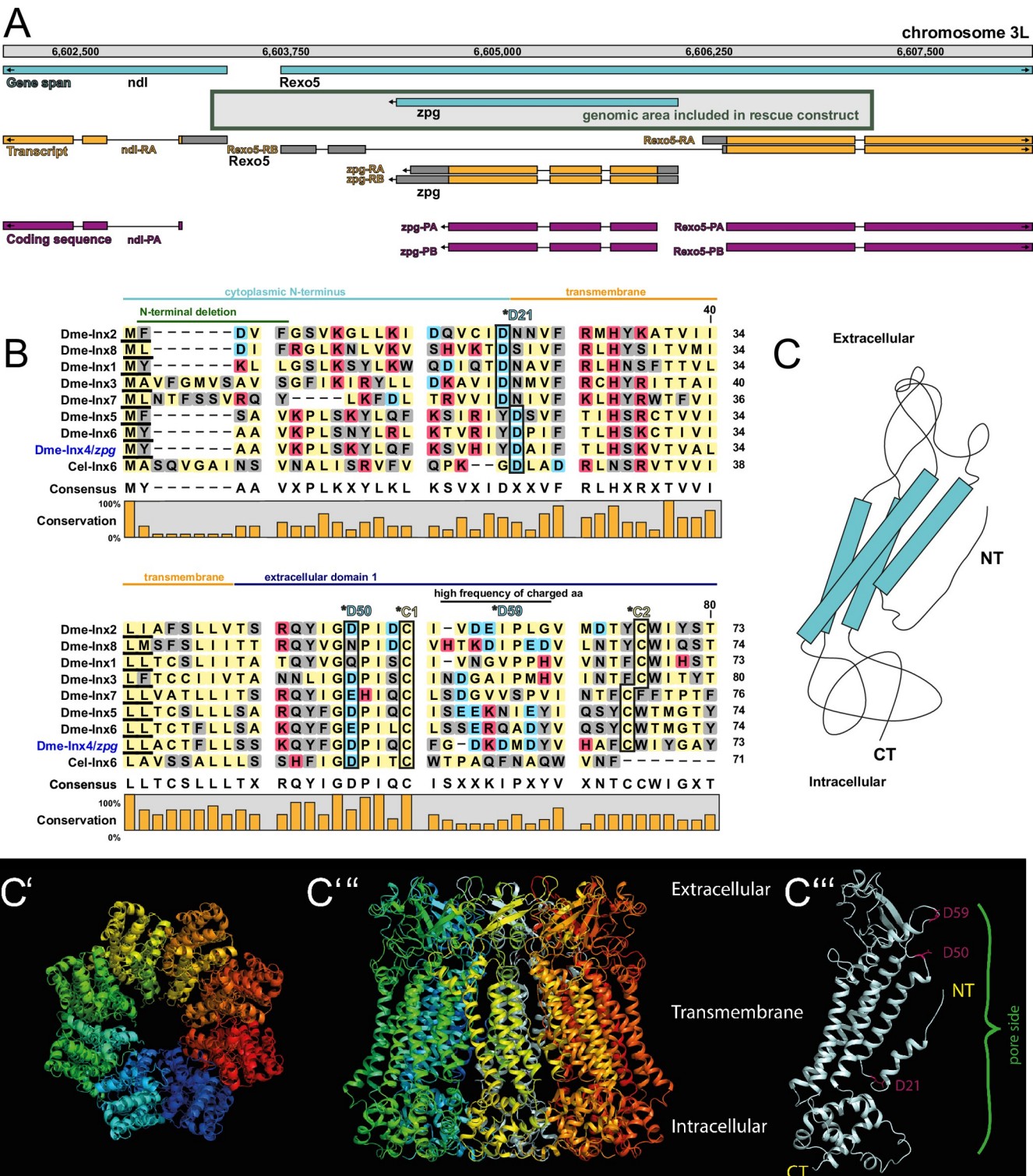

**Fig 1. The genomic locus of *zpg*, its protein structure, and the identification of residues of interest.** (A) Overview of the genetic locus of *zpg* (*inx4*) on chromosome 3L, showing genes (blue), transcripts (orange) and coding sequence (magenta) of Zpg and the neighboring genes. The DNA stretch that is included in the rescue construct used in this study is indicated by a grey box. (B) Sequence alignment of Zpg (Inx4) with other *Drosophila* innexins and *C. elegans* INX-6, which was used as basis for in silico 3D structure homology modeling. The N-terminal portions of the proteins are depicted (approximately amino acid 1–80, see numbers on the right) and the degree of conservation is indicated in bar graphs. Polar amino acids are shown in grey, hydrophobic in yellow, positively charged in magenta and negatively charged in cyan. The residues D21, D50 and D59 and well as the first (C1) and second cysteine (C2) in

Zpg, which were used as targets for mutagenesis, are part of this stretch of the protein and their location is indicated. Note that D21, D50 and the cysteine residues show a high degree of conservation among the innexins, whereas D59 does not. (C-C''') Predicted structure of *Drosophila* Zpg reveals octameric arrangement around a central pore. Simplified view in C. Top view in C'. Side view in C''. Each subunit is labeled in a different color. Single Zpg subunit is depicted in C''' and as a cartoon in C. The first extracellular domain as well as the entire N-terminus are facing inside the channel pore. Potentially functionally relevant residues within the channel opening are labeled in magenta (D21, D50, D59). While D21 and D50 are conserved among innexins, D59 was chosen as target for mutagenesis due to its predicted location at the narrow opening of the channel.

inside the pore and constrict its diameter. The putative structure of Zpg is depicted in Fig 1 (simplified cartoon in Fig 1C, top view in Fig 1C', side view in Fig 1C'', single subunit in Fig 1C'''). A color-coded version, shown in S1A Fig, was used to represent the per-residue score, with the most reliable positions in dark blue, intermediate in white, and least reliable in red. This indicated that the transmembrane region is the most reliable part of the model. The overall fold also agrees with an Alphafold2 model (S1B and S1C Fig). According to our modelling of Zpg, one channel is comprised of 8 subunits which link to each other to form a continuous round structure within the plasma membrane, leaving an open space between them that constitutes the channel pore. The transmembrane domains consist of highly parallel α-helices, whereas the other regions of the protein are less ordered. The C-terminus is fully intracellular, whereas the N-terminus (aa 1–21) is predicted to face inside the pore. As in INX-6, the extracellular domain 1 (E1, aa 43–110) is also partially located within the channel pore. Conserved residues within the channel pore are marked in magenta in Fig 1C'''. A simplified topological view is depicted in S2A Fig.

## Description of the methodology used for quantifying the level of rescue obtained by expressing different rescue constructs

Analysis of the rescue conferred by different mutated constructs requires a comprehensive quantitative measurement of the ability of the wildtype genomic *zpg*::GFP construct to ameliorate *zpg* mutant phenotypes. As shown previously by us and others [51–53], flies lacking *zpg* expression possess small rudimentary gonads and are sterile due to lack of germ cell differentiation and maintenance. When assessing rescue, four general aspects of testes structure and function were characterized, Zpg expression and localization, germ cell differentiation, soma development, and fertility.

First, Zpg expression was analyzed in wildtype (wt), *zpg* null mutant ($zpg^{z-2533}$ / $zpg^{z-5352}$) and *zpg*::GFP GR testes (a single copy of the *zpg*::GFP transgene introduced into the $zpg^{z-2533}$ / $zpg^{z-5352}$ background). In the wildtype, Zpg localizes to the soma-germline interface, outlining the developing cysts (Fig 2A and 2A'). In *zpg* null mutant testes, no Zpg staining can be detected, proving the specificity of the antibody (Fig 2B and 2B'). In *zpg*::GFP GR testes, Zpg distribution is identical to that seen in wildtype controls (Fig 2C and 2C'), though fluorescence intensity of Zpg staining is 40.8% lower compared to wildtype flies. The lower expression observed in *zpg*::GFP GR testes compared to wildtype flies is in itself not surprising since the *zpg*::GFP GR genotype, a *zpg* mutant rescued with one copy of genomic rescue construct, is functionally similar to heterozygous *zpg* mutants ($zpg^{2533}$/+). Consistent with this idea Zpg levels in heterozygous *zpg* was 34.4% lower compared to that seen in wildtype testes (S3 Fig). This is further supported by co-labelling, in a wildtype background, GFP, which tags the rescue construct (Fig 2D and 2D'), and Zpg. We previously showed that the C-terminal GFP tag blocks the epitope recognized by the Zpg antibody [53], making it possible to independently study the localization of either the GFP-tagged Zpg GR (using a GFP antibody, green) and the endogenous Zpg (using the Zpg antibody, red) and found that the GFP-tagged construct colocalized well with endogenous Zpg.

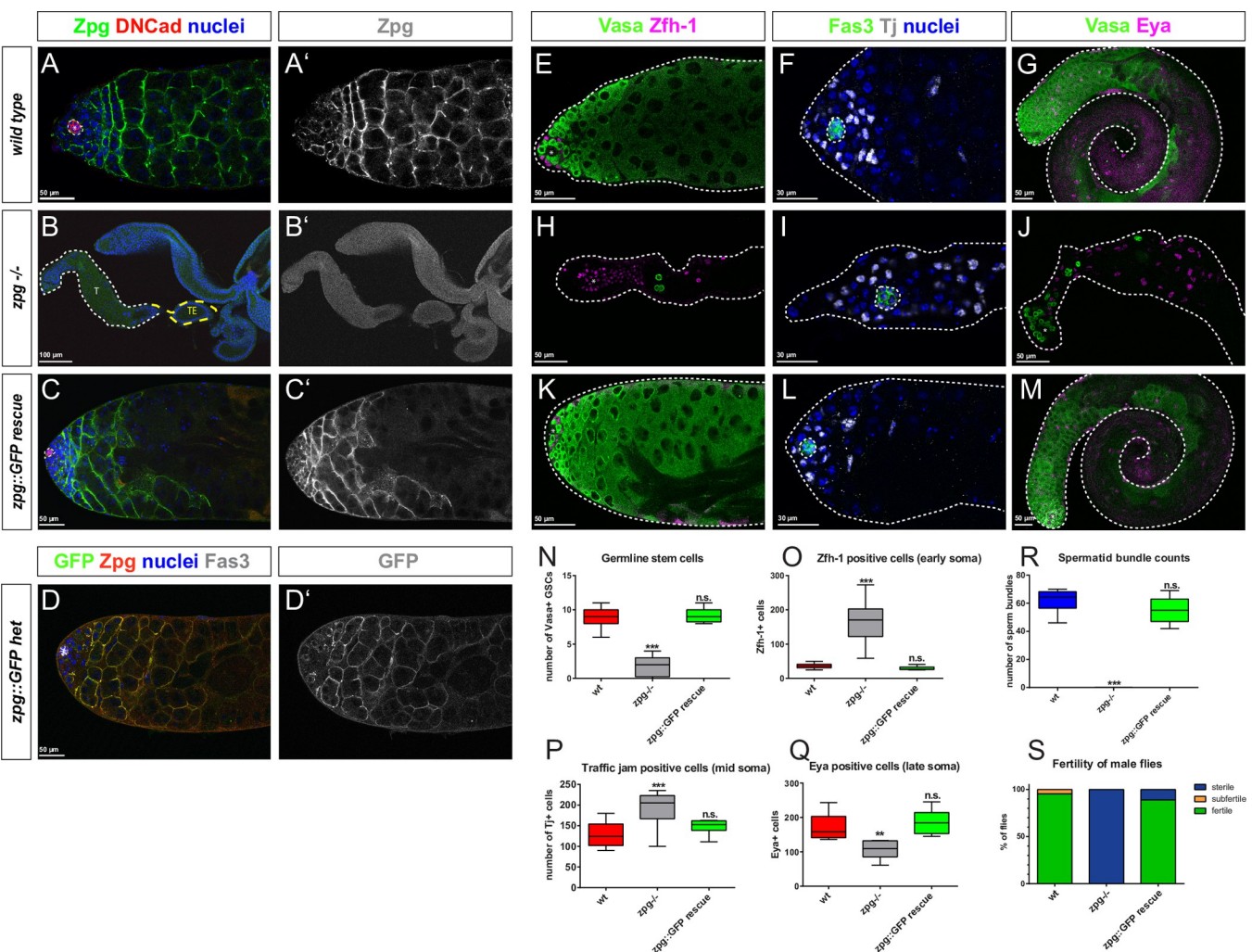

**Fig 2. Zpg function can be fully restored by introducing a GFP-tagged genomic rescue construct into the *zpg* null mutant background.** Zpg staining (green in A-C; single channels depicted in grey in A'-C') is strongly enriched at the soma-germline boundary in the wild type (A, A'), but cannot be detected in the rudimentary testes of *zpg* null mutants (B, B'). Wild type like distribution of Zpg can be seen in flies having the *zpg*::GFP GR (Genomic rescue) construct in the *zpg* null mutant background (C, C'). Hubs are marked by DN-Cadherin in red, nuclei are labeled in blue (A-C). Heterozygous expression of *zpg*::GFP GR (D, D') reveals strong colocalization of the transgenic construct (GFP, green) and endogenous Zpg (red). Fas3 labels the hub. D' shows single channel GFP signal at germ cell membranes in grey. Compared to wt (E, G), *zpg* null (H, J) mutants show a strong reduction of mitotic Vasa+ germ cells (green), indicating an early arrest in germ cell differentiation. In contrast, the number of germ cells (green) in *zpg*::GFP GR rescue flies (K, M) is indistinguishable from wt. The number of early somatic cells labeled by markers Zfh-1 (cyst stem cells and immediate daughter cells; magenta) and Tj (grey) is, compared to wt (E, F), greater in testes of *zpg* null mutants (H, I), but unaffected in *zpg*::GFP GR (K, L). Number of cells expressing the late somatic cell marker Eya (magenta) is, compared to wt (G), lower in *zpg* null mutants (J), but unaffected in *zpg*::GFP GR testes (M). Quantification of (N) the number of germline stem cells (GSCs, defined as single Vasa+ cells contacting the hub), (O) Zfh-1-positive cells, (P) Tj-positive cells, (Q) Eya-positive cells, (R) spermatid bundles, and (S) fertility in wildtype, *zpg* null mutant, and *zpg*::GFP GR rescue flies. Scale bars represent 30–100 μm, as indicated above them. p-values are for difference from wildtype and indicated by asterisks with *p<0.05, **p<0.01, ***p<0.001.

Second, germ cell differentiation was assessed by labelling germ cells and performing cell counts in wildtype, *zpg* null mutant, and *zpg*::GFP GR testes. The protein Vasa is a marker that labels germ cells from the germline stem cells (GSCs) stage to the spermatocyte stage [57]. In the wildtype (Fig 2E and 2G), Vasa mostly labels the anterior part of the testis, which harbors the mitotic-stage germ cells. In comparison, only a few Vasa expressing cells are seen in testes of mutants lacking Zpg expression (Fig 2H and 2J), which is in agreement with previously published data [51,53] and indicates a profound loss of germ cells in the mutants. Furthermore, in wildtype testes, both Vasa staining (Fig 2E and 2G) and Zpg staining (Fig 2A and 2A') stain the

developing cysts in which germ cells undergo incomplete mitotic divisions (cysts containing 2-, 4-, 8-, and 16- germline cells, respectively). In *zpg* null mutants (Fig 2H and 2J), no advanced cysts can be found, with cysts containing 2-germline cells being the most differentiated stage found in the majority of testes. In contrast to the *zpg* null mutants, the Vasa staining pattern in *zpg*::GFP GR testes (Fig 2K and 2M) strongly resembles the pattern observed in wildtype controls (Fig 2E and 2G), with many Vasa expressing cysts in the anterior part of the testis. GSCs are defined as single Vasa expressing cells that contact the hub. In the wildtype, on average 9.0 GSCs (n = 15) are found in the anterior tip of the testis. In *zpg* null mutants, however, the number of GSCs is significantly reduced to an average of 1.9 (n = 12; Fig 2N). The number of GSCs in *zpg*::GFP GR testes is nearly identical to the wildtype (average 9.2, n = 10; Fig 2N). Rescue was also assessed in late stage germ cells using the marker Boule, which specifically labels the meiotic germline [58]. In wildtype controls Boule expression is seen in spermatocytes and developing spermatids in the posterior part of the testis, with developing spermatid bundles appearing highly parallel (S4A Fig). In *zpg* mutants (S4C Fig), no Boule staining can be detected, as germ cells cannot reach meiotic stages. In *zpg*::GFP GR testes (S4B Fig) Boule expression is restored. In summary, these results show that germ cell differentiation is efficiently rescued in *zpg*::GFP GR flies.

Third, somatic cell differentiation was assessed by using markers to label distinct somatic populations and performing cell counts in wildtype, *zpg* null mutant, and *zpg*::GFP GR testes. To assess soma development, markers for different stages of somatic differentiation were used. In particular, we analyzed the expression of the early soma marker Zinc Finger Homeodomain-1 (Zfh-1), the intermediate soma marker Traffic Jam (Tj), and the late somatic marker Eyes Absent (Eya). In wildtype controls, Zfh-1 is expressed in a small population of 36 cells (on average, n = 18) in close proximity to the hub (Fig 2E). Cell types labelled by Zfh-1 include CySCs and their immediate daughter cells [59]. Moreover, Tj typically marks a population of 131 cells (on average, n = 20) also in proximity to the hub (Fig 2F; [60]). Finally, the late somatic cell marker Eya [61] (Fig 2G) labels a population of 174 cells (on average, n = 8) distributed throughout the testis. In *zpg* mutants, the average number of both Zfh-1 and Tj positive cells increased (Fig 2H and 2I, quantified in Fig 2O and 2P; mean 165, n = 20; and 194, n = 18 for Zfh-1 and Tj, respectively) In contrast, the average number of Eya expressing cells was reduced in *zpg* mutant testes (Fig 2J; quantified in Fig 2Q; mean 107, n = 8). These numbers are in line with previously published results [53] and demonstrate the misregulation of somatic cell differentiation, with an increase in the population of early somatic cells and a reduction in the size of the late somatic population in the *zpg* null mutants. In *zpg*::GFP GR flies, the somatic cell counts for Zfh-1 (Fig 2K, quantified in Fig 2O; mean 30, n = 18), Tj (Fig 2L; quantified in Fig 2P; mean 148, n = 8) and Eya (Fig 2M; quantified in Fig 2Q; mean 189, n = 8) expressing cells are very similar to those seen in wildtype controls (Fig 2G). In summary, these results show that somatic cell differentiation is not impaired in *zpg*::GFP GR flies.

Fourth, flies lacking Zpg expression are unable to produce sperm, rendering male flies sterile [51]. Spermatid bundles appear as arrowhead shaped structures strongly stained with DAPI (S4F Fig), which are mostly localized within the posterior part of the testis. In the wildtype, on average 62 sperm bundles can be detected (Fig 2R, n = 10), whereas *zpg* mutants fail to produce sperm altogether. Zpg::GFP GR testes appeared to have a slightly lower number of spermatid bundles compared to wildtype controls (Fig 2R, average 55, n = 20), however this reduction was not statistically significant and did not have any influence on the fertility of the flies (Fig 2S). About 95% of tested wildtype males (n = 44) were fully fertile compared to 88% of *zpg*::GFP GR flies (n = 36), whilst none of the tested *zpg* null mutant males (n = 54) were able to produce offspring (Fig 2S). Taken together, these results show that the zpg::GFP construct can effectively rescue the *zpg* null mutant phenotype, both in the germline and the soma, demonstrating the efficiency of our rescue approach.

## The C-terminal domain is essential for Zpg localization

To generate mutant lines for the structure/function analysis we inserted mutations in the *Zpg*::GFP construct and generated transgenes containing these modified rescue constructs (see Materials and Methods; S2B–S2B''' Fig). To ensure uniform expression, all transgenes used in this study, including the wildtype rescue construct, were inserted into the same chromosomal location using the φC31-based integration system [62]; see materials and methods). The C-terminal cytoplasmic domains of gap junction proteins are known for their many important functions [35]. Loss of the C-terminus of Cx43 leads to lethality due to a defective epidermal barrier in mice [63], and to oculodentodigital dysplasia in humans [64]. Therefore the role of the C-terminus domain of Zpg was analyzed by generating a mutant line in which the C-terminus was replaced by a GFP tag (*zpg* deltaCT::GFP; see Materials and Methods), and studying its ability to rescue *zpg* mutants. Overall, testes of *zpg* deltaCT::GFP mutants appeared rudimentary (Fig 3B, 3B' and 3J–3L) and resembled the testes of *zpg* null mutants (compare to Fig 2). As expected, since our Zpg specific antibody recognizes the C-terminus [53] no Zpg antibody staining could be detected in *zpg* deltaCT::GFP mutants (Fig 3B and 3B'), illustrating the specificity of the antibody. The GFP tag, however, enables detection of the C-terminal deletion construct in the *zpg* null mutant background (Fig 3E and 3E') and the GFP signal was seen within the germ cells where it appeared to be mostly cytoplasmic. In heterozygous flies having one copy of *zpg* deltaCT::GFP and one copy of endogenous *zpg* (Fig 3F and 3F'), the GFP-tagged protein was located intracellularly, where it accumulated in the cytoplasm, whereas the endogenous Zpg (marked by the Zpg antibody) localized to the soma-germline interface as expected. This shows that the deltaCT::GFP transgene is expressed, but is not capable of localizing to the plasma membrane, indicating that the C-terminus is required for trafficking of Zpg to, and/or maintenance of Zpg at, the plasma membrane. As a consequence of the inability of the Zpg deltaCT::GFP protein to localize, severe defects in germ cell and somatic cell differentiation were observed when it was used to rescue *zpg* mutants. Few early germ cells were observed in the testes of *zpg* deltaCT::GFP mutants (Fig 3J and 3L) compared to wildtype controls (Fig 3G and 3I) and germ cell differentiation was arrested before or at the 4 cell cyst stage (Fig 3G and 3I for wt,3J, 3L for mutant). Quantification of the number of GSCs in *zpg* deltaCT::GFP mutants (Fig 3S; mean 3.4, n = 11) showed a significant loss of stem cells compared to wildtype, but a slightly higher number compared to *zpg* null mutants. Since germ cell development was arrested early on in *zpg* deltaCT::GFP mutants, the germ line did not differentiate to the meiotic stages. Similarly, somatic cell phenotypes in *zpg* deltaCT::GFP mutants resembled those seen in *zpg* null mutants (Fig 3T–3V). In *zpg* deltaCT::GFP mutants, the number of cells stained with the early somatic cell markers Zfh-1 and Tj was increased compared to wildtype (Fig 3G, 3H, 3J, 3L, 3T and 3U; mean of 162.2 and 202.3, n = 9 and 7, respectively). In contrast, the number of cells expressing the late somatic cell marker Eya was reduced compared to wildtype to a similar extent to that seen in *zpg* null mutants (Fig 3J, 3L and 3U; mean 88.3, n = 8). Finally, no spermatid bundles were found in *zpg* deltaCT::GFP mutants (Fig 3W), resulting in sterility (Fig 3X, n = 19). In summary, our data shows that the loss of the C-terminus disrupts Zpg delivery to, and/or maintenance at, the membrane, and that C-terminal deletion mutants give rise to null-like phenotypes.

## Conserved phosphorylation sites in the C-terminal domain are not essential for Zpg function

Phosphorylation of key conserved residues in the C-terminal domain of gap junction proteins is known to regulate their assembly, turnover, channel conductance, and cargo selectivity [35,65,66]. Zpg has only two conserved putative phosphorylation sites in its C-terminus, Y352

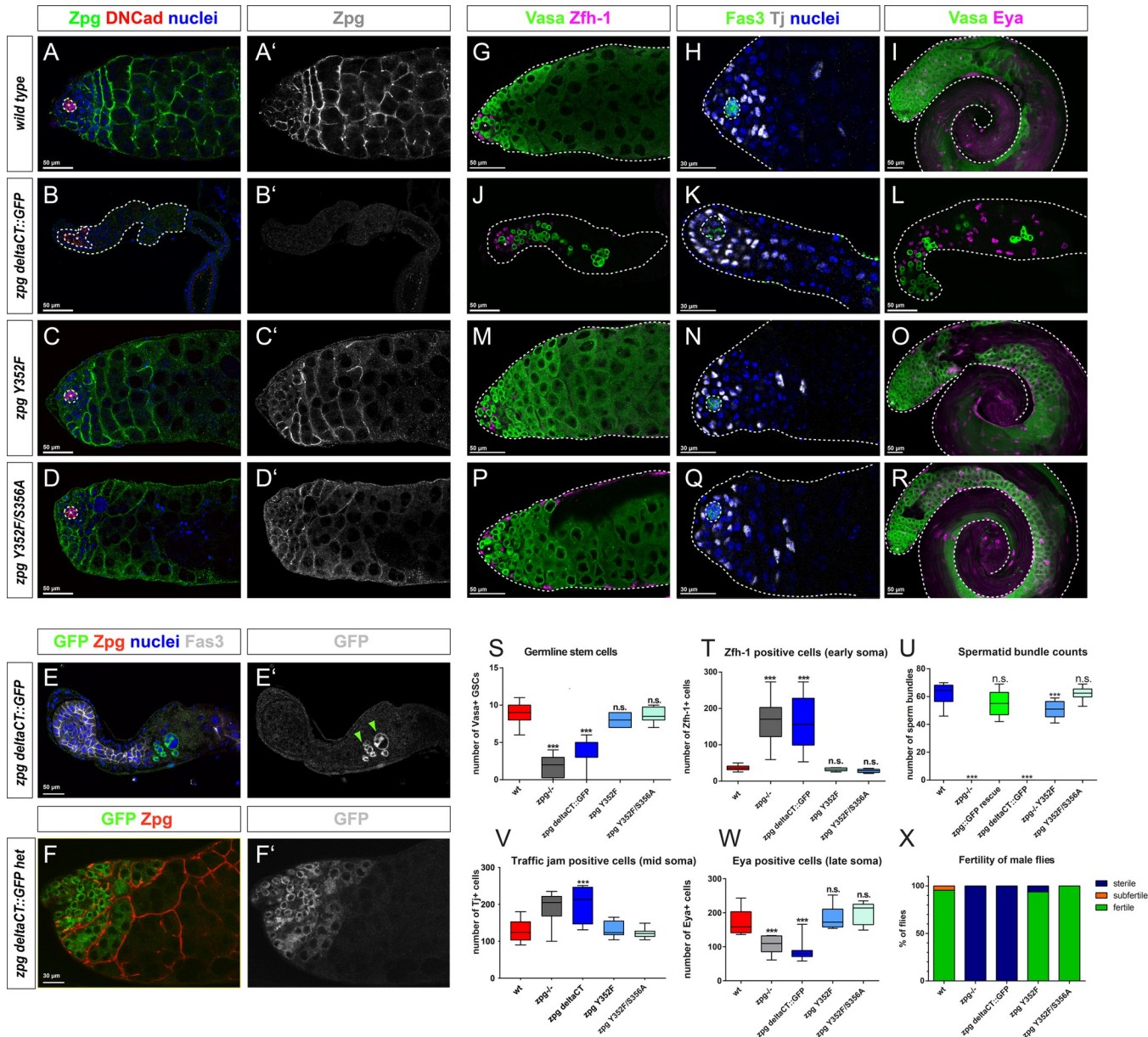

**Fig 3. The C-terminus of Zpg is crucial for protein function, while C-terminal phosphorylation sites are dispensable.** Zpg staining (green in A-D; single channels in grey in A'-D') is absent in *zpg* mutants rescued with Zpg containing a C-terminal deletion (*zpg* deltaCT::GFP) (B, B'), as the antibody binding site is deleted. *zpg* deltaCT::GFP mutant testes are severely reduced in size and the hub (DN-Cadherin, red) is enlarged. In *zpg* mutants expressing a genomic rescue construct containing mutations in phosphorylation sites (C, C': *zpg* Y352F, D, D': *zpg* Y352F/S356A), Zpg staining normally localizes to the germline-soma boundaries (as indicated by arrows). Wt control is shown in A, A'. Nuclei are highlighted in blue. (E-F) localization of the GFP tag in testes when *zpg* deltaCT:: GFP is expressed in the null mutant background (E-E') and in flies with one copy of endogenous *zpg* (F-F'). In testes of both genotypes, the GFP signal accumulates intracellularly. Compared to wt (G, I), significantly less Vasa+ early germ cells (green) can be detected in testes of *zpg* deltaCT::GFP flies (J, L), whereas no difference to wt was seen in *zpg* Y352F (M, O) and *zpg* Y352F/S356A mutants (P, R). The number of Zfh-1+ cells (wt shown in F) and Tj+ cells (wt shown in H) was higher in *zpg* deltaCT::GFP testes (J, K), but not in the phospho mutants (M-Q). Less cells expressing the late somatic marker Eya were detected in testes of *zpg* deltaCT::GFP flies (L) than in wt (I), but no change was found in the phosphorylation mutants (O, R). This indicates defective germ cell and somatic cell differentiation in *zpg* deltaCT::GFP flies, but not in the two phosphorylation mutants. Quantification of (S) germline stem cells (GSCs), (T) Zfh-1-positive cells, (U) Tj-positive cells, (V) Eya-positive cells, (W) spermatid bundles, and (X) fertility, shows loss of function upon deletion of the C-terminus, but no defects in phosphorylation mutants. Scale bars represent 30–50 μm, as indicated above them. p-values are for difference from wildtype and indicated by asterisks with $^*p<0.05$, $^{**}p<0.01$, $^{***}p<0.001$.

and S356, and both were mutated to determine their contribution to Zpg function. Two different mutants were generated: *zpg* Y352F and a double mutant (*zpg* Y352F/S356A) that knocks out both of the conserved putative phosphorylation sites. Both Zpg Y352F (Fig 3C and 3C') and Zpg Y352F/S356A (Fig 3D and 3D') could localize to the membrane of germ cells but their localization was somewhat patchy and less contiguous compared to wildtype controls. Vasa staining in *zpg* Y352F (Fig 3K and 3O) and *zpg* Y352F/S356A mutants (Fig 3P and 3R) was indistinguishable from wildtype controls (Fig 3G and 3I), with an abundance of large Vasa expressing cysts in the anterior part of the testis, indicating that germ cell differentiation was not impaired. Accordingly, the number of GSCs was not altered in either Y352F or Y352F/S356A mutants (Fig 3S; averages of 8.0 and 8.6 GSCs/testis, n = 9 and 8, for the Y352F, and Y352F/S356A mutants, respectively). Moreover, Boule staining (S4D and S4E Fig) was similar to that seen in wildtype controls for both *zpg* Y352F and Y352F/S356A mutants, meaning that germ cells entered meiosis normally. Similarly, somatic differentiation was not perturbed in the *zpg* Y352F and Y352F/S356A mutants compared to wildtype controls (Fig 3T–3V). Specifically, the number and distribution of cells expressing the early somatic markers Zfh-1 (Fig 3M and 3P, quantified in Fig 3T; mean of 31 cells/testis for Y352F and 28 cells/testis for Y352F/ S356A, n = 8 for each genotype) and Tj (Fig 3N and 3Q, quantified in Fig 3U; mean of 131.6 cells/testis for Y352F and 122.3 cells/testis for Y352F/S356A, n = 8 and 9, respectively), as well as the late somatic cell marker Eya (Fig 3O and 3R, quantified in Fig 3T; means of 186 cells/testis for Y352F and 202 cells/testis for Y352F/S356A, n = 8 and 7, respectively), were all indistinguishable from wildtype controls (Fig 3I). Consistent with these observations the number of spermatid bundles was wildtype for the Y352F/S356A mutant (Fig 3W, 62.4 vs. 62 in wt, n = 10 for each genotype), near wildtype for the *zpg* Y352F mutant (avg. of 50.7 spermatid bundles, n = 10), and both *zpg* Y352F (n = 33) and the Y352F/S356A mutants (n = 47) were fully fertile (Fig 3X). Overall, we did not detect any meaningful phenotypic defects or reduced fertility in the phosphorylation site mutants.

## Zpg function requires coupling to other GJ proteins in neighboring cells

The vertebrate homologs of the innexins are pannexins, which are known to predominantly function as hemichannels, rather than cell to cell channels, enabling the passage of cargo between the cytoplasm and the extracellular space [24]. It is currently unclear how much of innexin function, if any, can be due their capacity to form hemichannels versus gap junctions and we used our rescue methodology to explore this question. The formation of gap junctions requires the docking of a Connexin or Innexin multimeric hemichannel (so called Connexons or Innexons) to another hemichannel in a neighboring cell via a set of extracellular cysteine surface residue, an interaction that is mediated by disulfide bridges [22, 23]. The Zpg protein has a clearly defined set of 6 surface cysteine residues in its extracellular loops, identified by structure, location, and sequence conservation, that can mediate these disulfide bridges. Exchanging the cysteine residues to other amino acids would disrupt the ability of Zpg to form disulfide bridges with an innexin present of the surface of adjacent somatic cells. We generated three *zpg* mutants in which different cysteine residues were replaced with serine residues. Specifically, and following the convention of numbering the 6 surface cysteine residues of Zpg from 1 to 6 starting at the C-terminal end, we generated the mutants (see Materials and Methods): *zpg* C6S (6th cysteine mutated to serine), *zpg* C145S (1st, 4th and 5th cysteine mutated to serine), and *zpg* C236S (2nd, 3rd and 6th cysteine mutated to serine). Initial assessment of the testes in the three mutant lines showed a significant size reduction compared to wildtype controls (Fig 4A–4D). Zpg was not observed at the cell membrane in germ cells in the Cysteine mutants (Fig 4A'–4D') but instead appeared as diffuse cytoplasmic specks. In order to better

visualize the localization of Zpg upon mutation of cysteine residues, we generated two GFP-tagged cysteine mutant fly lines, C6S::GFP ($6^{th}$ cysteine replaced by serine) and C26S::GFP ($2^{nd}$ and $6^{th}$ cysteine replaced by serine). In the *zpg* null mutant background the localization of Zpg CS6::GFP and Zpg C26S::GFP in germ cells also appeared cytoplasmic. We again took advantage of the fact that the C-terminal GFP tag blocks the epitope recognized by the Zpg antibody [53], and using the GFP antibody (green) combined with the antibody staining the endogenous Zpg (red) (S5Q–S5T Fig). These experiments were carried out in heterozygous flies containing one copy of the mutated Zpg CS6::GFP or Zpg C26S::GFP, respectively, and one copy of the endogenous Zpg. We observed low levels of colocalization, measured by calculating the Pearson colocalization coefficient, between the endogenous and the mutated Zpg (average r = 0.41 in both *zpg* CS6::GFP (n = 11) and *zpg* C26S::GFP (n = 14) compared to the average r = 0.8 (n = 11) in *zpg*::GFP GR, S5U and S9 Figs). Specifically, the mutated proteins were expressed but remained cytoplasmic. This suggests a possible role for the surface cysteine residues in membrane localization but could also indicate possible issues with protein stability, though, being surface residues, mutations in the surface cysteines are unlikely to impact protein folding or packing.

In line with the small testis size and absence of Zpg at the plasma membrane, fewer, and seemingly undifferentiated germ cells, were observed in all three Cysteine mutants (Fig 4I and 4K for *zpg* C6S, 4L, 4N for *zpg* C145S, 4O, 4Q for *zpg* C236S). Quantification of GSC numbers revealed a significant loss of stem cells in all cysteine mutants (Fig 4R, with wt average of GSCs per testis at 9 (n = 15), compared to 2.6 in *zpg* C6S (n = 8), 3.1 in *zpg* C145S (n = 8), 2.1 in *zpg* C236S (n = 9)). The severe germ cell differentiation defect meant that the germline in *zpg* cysteine mutant testes did not reach the meiotic stages. In all three *zpg* cysteine mutant fly lines, somatic cell differentiation defects manifest in a similar way as in *zpg* null mutants. Cells staining positive for the early somatic cell markers Zfh-1 (Fig 4F, 4I, 4L and 4O, quantified in Fig 4S; means of 178.4 for C6S, 127.5 for C145S and 171.3 for C236S compared to 35.9 in wt, n = 9, 8, 8, 20 respectively) and Tj (Fig 4G, 4J, 4M and 4P, quantified in Fig 4T; means of 193.4 for C6S, 190.4 for C145S and 182.7 for C236S compared to 130.6 in wt (Fig 4G) n = 10, 13, 11, 20 respectively) were significantly more abundant than in wildtype controls. The late somatic cell marker Eya, in contrast, was found to label significantly less cells in the cysteine mutants than in wildtype controls (Fig 4H, 4K, 4N and 4Q, quantified in Fig 4U; means 89.6 for C6S (n = 7), 92.6 for C145S (n = 8) and 94.1 for C236S (n = 8) compared to 173.7 in wt (n = 8)). Moreover, no spermatid bundles were found in any of the three cysteine mutant fly lines (Fig 4V, n = 10 per fly line), ultimately resulting in complete sterility (Fig 4W, n> 44 per fly line). For all three cysteine mutant lines, the hubs appeared enlarged compared to wildtype controls, and even in comparison to *zpg* null (Fig 4A–4D). In support of this analysis, similar overall phenotypes were obtained in analysis of the GFP tagged version of CS6 or C26S mutants (*zpg* CS6::GFP or *zpg* C26S::GFP; see S5 Fig). We conclude that the *zpg* cysteine mutant behave, for the most part, indistinguishably from null alleles of *zpg* suggesting that Zpg activity requires an ability to form viable gap junctions, with the mutation of even a single disulfide bridge-forming cysteine residue leading to complete loss of Zpg function.

### Identification and mutagenesis of key residues within the Zpg channel pore that modulate the interaction with cargo and pore configuration

Gap junctions can allow the passage of many different cargos, to determine the role of cargo specificity in gap junction-mediated communication in the testes we set out to generate mutations that would modulate the passage of various cargos without blocking channel function. It has been proposed that the N-termini of GJ proteins, which can reside inside the channel pore,

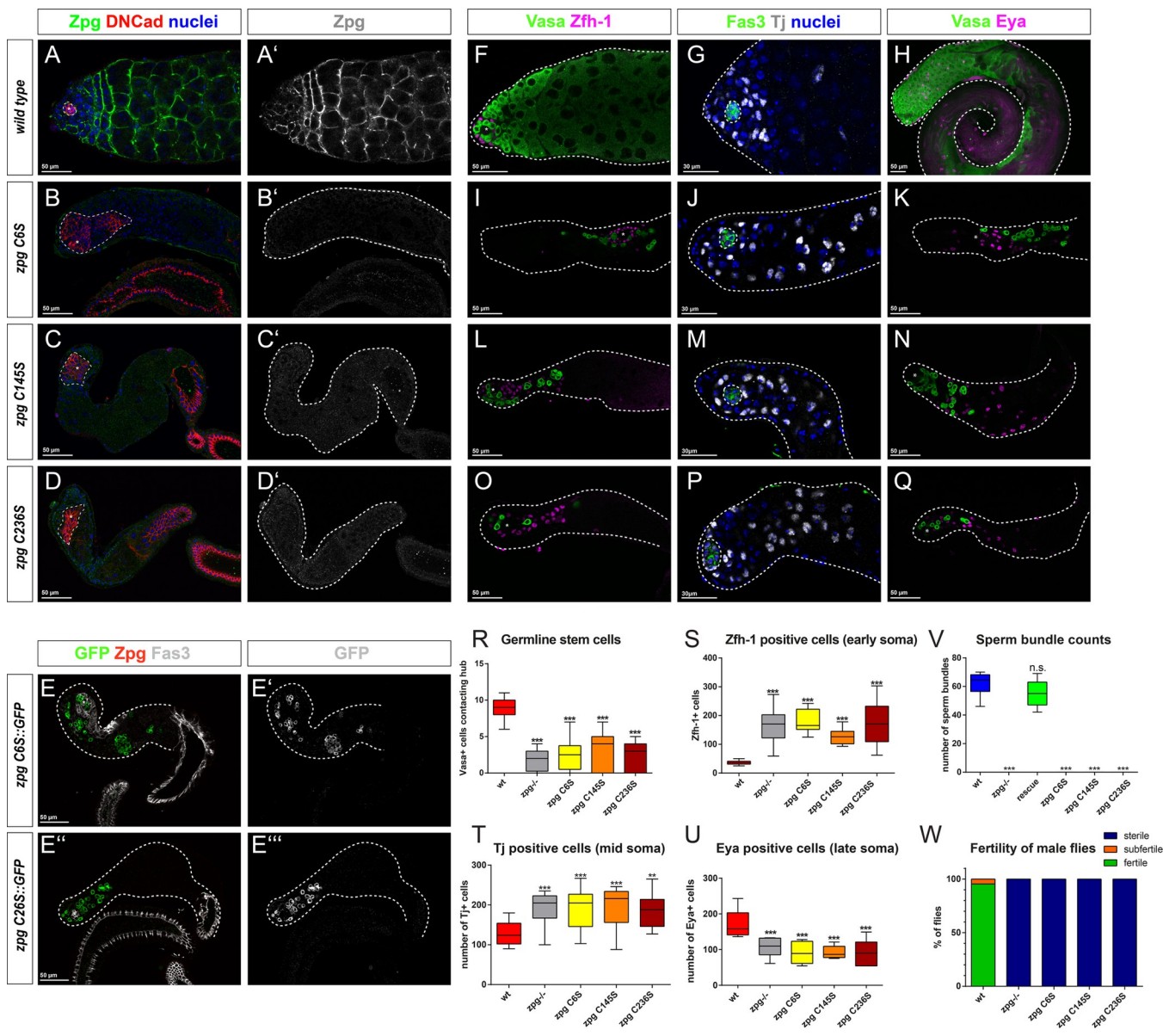

**Fig 4. Zpg does not function as a hemichannel.** (A-D) *zpg* mutants rescued with genomic rescue constructs in which one or more cysteine residues were mutated, hindering the formation of gap junctions, have rudimentary testes and no Zpg is detected by antibody staining (green in A-D; single channels depicted in grey in A'-D'; wt in A-A'; *zpg* C6S, B-B':, *zpg* C145S, C-C'; *zpg* C236S, D-D'). Hubs are marked with DNCad in red, nuclei are highlighted in blue. (E-E''') In testes of *zpg* mutants expressing GFP-tagged versions of the cysteine mutation constructs (*zpg* C6S::GFP in E, E', *zpg* C26S::GFP in E'', E'''), a strong intracellular accumulation of the GFP signal can be detected, while Zpg antibody staining is very weak. Cysteine mutations in *zpg* cause a strong defect in early stages of germ cell differentiation as detected by Vasa staining (F, H: wt; I, K: *zpg* C6S, L, N: *zpg* C145S; O, Q: *zpg* C236S). Compared to wt (F, G), the expression of the early somatic markers Zfh-1 (magenta; I, L, O) and Tj (grey; J, M, P) was increased in all three cysteine mutants, whereas the number of cells expressing the late marker Eya (K, N, Q) was decreased. Quantification of (R) germline stem cells (GSCs), (S) Zfh-1-positive cells, (T) Tj-positive cells, (U) Eya-positive cells, (V) spermatid bundles, and (W) fertility shows null mutant-like phenotypes in cysteine mutants, leading to complete sterility. Scale bars represent 30–50 μm, as indicated above them. p-values are for difference from wildtype and indicated by asterisks with *p<0.05, **p<0.01, ***p<0.001.

most likely play a role in channel gating and selectivity [55,56,67]. To determine if the N-terminal of Zpg could play such a role, we utilized the *C. elegans* INX-6 based homology-model (Fig 1). The model suggested that both the N-terminus (NT) as well as first stretch of the Extracellular Domain 1 (E1) face inside the channel pore. Moreover, alignment of *Drosophila*

innexin amino acid sequences (Fig 1B) identified three conserved aspartate residues, which are positioned in key locations within the channel pore: D50 (see Fig 5A), D59 (see Fig 6A), and D21 (see Fig 7A). The highly conserved aspartate 21 (D21) residue was located at the end of the N-terminal helix inside the pore. Although the exact side chain conformation cannot be unambiguously assigned from homology modeling, its overall location near the pore suggests that it may interact with cargo that passes through the pore (Fig 7A'). A second aspartate at position 50 (D50) is found close to the narrowest constriction of the pore (Fig 5A). Although this residue is only partially conserved between innexins and connexins, it is found in a number of connexins, and a mutation in the D50 residue in human Cx26 (D50N) is implicated in keratitis-ichthyosis-deafness (KID) syndrome (Sanchez et al., 2013) [68]. Our homology model suggests that D50 may interact with glutamine 46 (Q46) of the adjacent subunit of Zpg (Fig 5A'), thereby contributing to the conformation and stability of the pore. A third conserved aspartate at position 59 (D59) is also situated at the narrowest part of the channel pore, constricting its diameter (Fig 6A for top view). This residue is also near an intersubunit interface, and may make interactions with lysine 58 (K58) of a neighboring subunit of Zpg (Fig 6A'). Due to its location near the narrowest part of the channel, the interactions mediated by D59 may affect the structural configuration of the pore and contribute to cargo selectivity.

Although the precise interactions of these three Asp residues will have to await experimental verification, their overall locations suggest they play important roles in channel function. We therefore further explored their role using site-directed mutagenesis. Three different types of point mutations were introduced in the D21, D50 and D59 residues to alter amino acid polarity, modify their interaction with positively charged cargo, and change the nature of hydrogen bonds that can be formed, respectively. First, as D is a polar and negatively charged amino acid, replacing it with a polar, positively charged amino acid (arginine (R), lysine (K), or histidine (H)) directly reverses the charge. While these positively charged residues could still form hydrogen bonds, this would be with a different residue, thereby altering channel conformation. Second, as alanine (A) is a hydrophobic amino acid, introducing a D to A point mutation impinges on the formation of hydrogen bonds or, in the case of D21, the interaction with positively charged ions. Therefore, D to A mutations would be predicted to be the most significant functional change within our mutagenesis approach. Third, a milder type of mutation was introduced by mutating D to asparagine (N). Such a change to a polar but uncharged residue is predicted to modulate the strength and nature of hydrogen bonds that can form in the pore. In addition to these point mutations, a more drastic mutation was generated in which the first four amino acids of Zpg excluding the methionine were deleted (*zpg* delta2-5). Schematic models of the residues that were targeted in our mutagenesis are depicted in Figs 5A", 6A" and 7A".

## Channel pore mutations in Zpg localized to the Plasma Membrane

To study the localization of channel pore mutants they were GFP-tagged at their C-terminus. Since the GFP fusion at the C-terminal prevents recognition by the Zpg antibody this allows us to distinguish the localization of the Zpg encoded by the rescue construct (GFP positive and Zpg antibody negative) from the endogenous Zpg (GFP negative but Zpg antibody positive). Using this approach, we observed normal membrane localization of the Zpg protein encoded by the rescue construct containing either of 3 different mutations in the D50 residue, D50A, D50R or D50K (see Materials and Methods, Figs 5B–5F, S6, Table 1). Similarly, the Zpg protein encoded by the rescue construct containing either of 3 different mutations in the D59 residue, D59A, D59N, or D59H localized normally to the surface of germ cell (Figs 6B–6F, S7, Table 1). In comparison, the Zpg protein encoded by the rescue construct containing either of

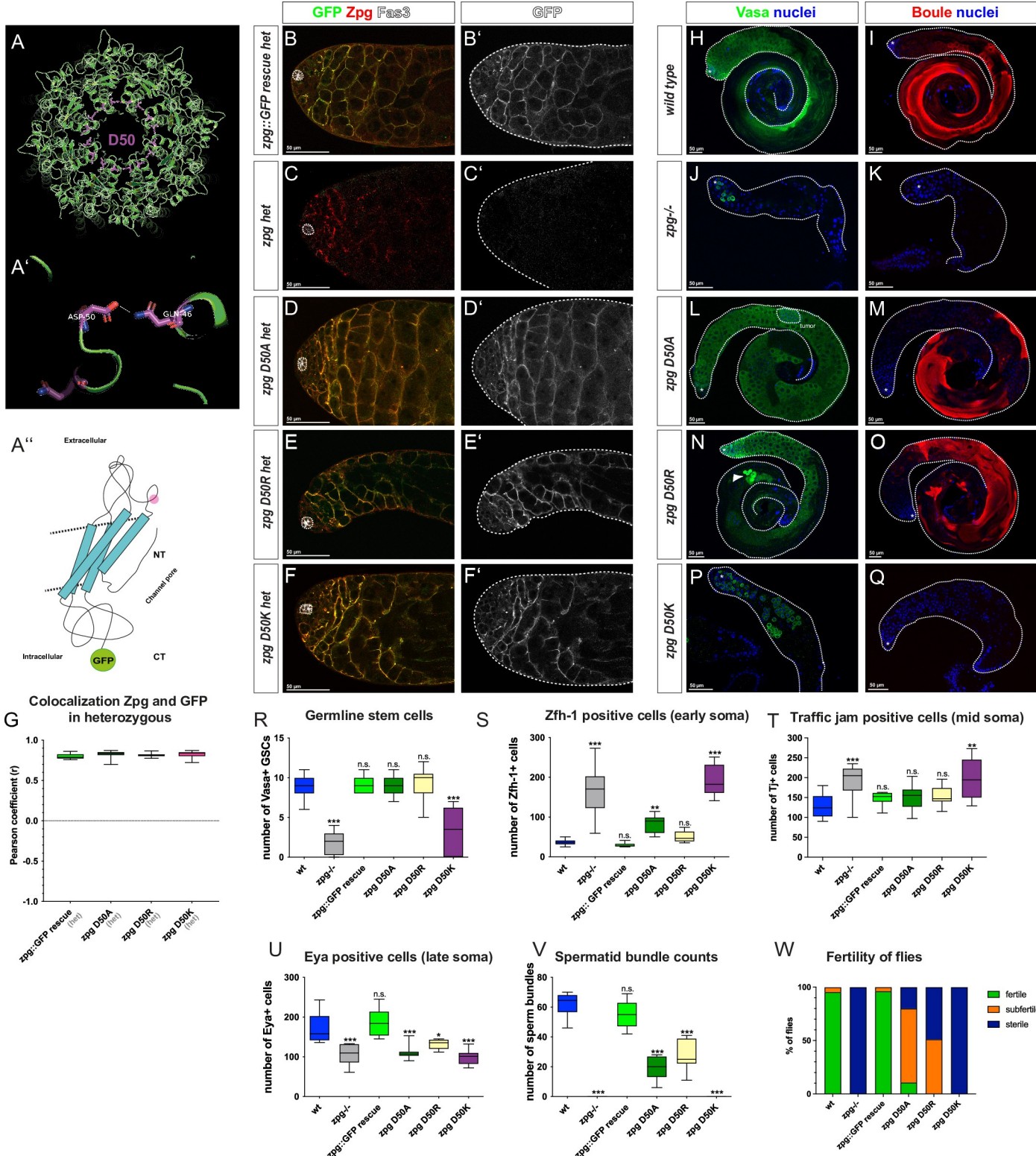

**Fig 5. Mutations in the Zpg D50 channel pore residue result in germ cell differentiation defects.** (A) Homology model of Zpg showing the position of aspartate 50 (D50) within the channel pore. (A') D50 is predicted to form a hydrogen bond with glutamine 49 (Q49) of the adjacent Zpg subunit. (A''') Simplified model highlighting the location of D50 (marked in pink) in the first extracellular loop. (B-F) Colocalization of wildtype endogenous Zpg and GFP-tagged, mutated Zpg. Flies heterozygous for a null allele of zpg but also containing one copy of the wildtype genomic *zpg* rescue construct (*zpg* GFP::GR; B-B'), no rescue construct (C-C'),

one copy of the *zpg* D50A mutant rescue construct (D-D'), one copy of the *zpg* D50R mutant rescue construct (E-E') and one copy of the *zpg* D50K mutant rescue construct (F-F'), the GFP-tagged D50 mutants show strong colocalization with the endogenous Zpg at the membrane. This high degree of colocalization is also revealed by the quantification of the Pearson coefficient between the GFP and Zpg antibody staining (G). (H-Q) Staining for the mitotic germ cell marker Vasa and the late-stage germ cell marker Boule. In the wildtype, Vasa staining is mostly concentrated in the apical part of the testis (H), whereas Boule marks meiotic cysts and long, parallel bundles of spermatids (I). In *zpg* null mutant testes little Vasa (J) and no Boule (K) signal is detected. In both *zpg* D50A (L) and *zpg* D50R (N) testes, the Vasa signal is strong and broadly localized. However, in *zpg* D50A testes, Vasa-positive cysts are abnormally found throughout the entire testis (L), and defective cysts can be observed in both mutants (circled in L, arrowhead in N). In addition, Boule staining in testes of *zpg* D50A (M) and *zpg* D50R (O) mutants reveals disorganized spermatid bundles. While *zpg* D50K mutant testes have a larger number of germ cells and larger mitotic cysts compared to *zpg* null mutants (P), they fail to reach meiosis (Q). Quantification of (R) germline stem cells (GSCs), (S) Zfh-1-positive cells, (T) Tj-positive cells, (U) Eya-positive cells, (V) spermatid bundles, and (W) fertility data. The data indicates late germ cell differentiation defects in *zpg* D50A and *zpg* D50R mutants and a stronger phenotype closer to the null mutant in *zpg* D50K mutants. Hubs are either encircled or indicated by asterisks. Scale bars represent 50 μm, as indicated above them. n>30 single crosses per genotype for fertility tests. p-values are for difference from wildtype and indicated by asterisks with $^*p<0.05$, $^{**}p<0.01$, $^{***}p<0.001$.

2 different mutations in the D21 residue, D21A or D21N, showed substantial cytoplasmic localization though some protein was able to localize to the plasma membrane (Figs 7B–7F, S8, Table 1). Finally, the Zpg protein encoded by the rescue construct containing the N-terminal delta2-5 truncation (Figs 7F, and S8, Table 1) also exhibited a wildtype pattern of localization to the plasma membrane.

Quantifications of the Pearson colocalization coefficient between the endogenous and the mutated Zpg also reveals weak colocalization in both *zpg* D21 mutants (average r = 0.44 in *zpg* D21A (n = 11) and r = 0.58 in *zpg* D21N (n = 12) compared to average r = 0.8 (n = 11) in *zpg*::GFP GR (Figs 7G and S9), whereas all other mutant constructs showed very strong colocalization with endogenous Zpg (average r = 0.82 in *zpg* D50A (n = 10), r = 0.81 in *zpg* D50R (n = 10), r = 0.82 in *zpg* D50K (n = 12), r = 0.81 in *zpg* D59A (n = 10), r = 0.75 in *zpg* D59H (n = 10), r = 0.84 in *zpg* D59N (n = 10) and r = 0.84 *zpg* delta2-5 (n = 12) compared to average r = 0.8 (n = 11) in *zpg*::GFP GR (Figs 5G, 6G and 7G and S9). These results show that Zpg proteins containing mutations in the channel pores were, for the most part, able to stably localize to the plasma membrane.

## Channel pore mutations in Zpg exhibit a range of phenotypes

In terms of general testes morphology and overall fertility, a range of phenotypes was seen when channel pore mutations were used to rescue *zpg* null flies (Table 1). Testes from both the *zpg* D50A and *zpg* D50R mutants appeared morphologically wildtype if slightly smaller than controls, produced spermatid bundles, and were fertile, though less fertile than wildtype controls (Fig 5H and 5I for wt, 5L–5O for mutants, 5V, 5W). In comparison, the *zpg* D50K mutants had rudimentary testes that resembled those of *zpg* null flies (Fig 5J and 5K for null, 5P, 5Q for D50K, 5V, 5W), no spermatid bundles were produced, and flies were sterile. Testes from *zpg* D59N, and *zpg* D59H mutants also appeared morphologically wildtype but exhibited a smaller size, lower number of spermatid bundles, and reduced fertility compared to the controls (Fig 6H and 6I for wt, 6N–6Q for mutants, 6V, 6W). A stronger reduction in size, and severe reduction in both the number of spermatid bundles and fertility, was seen in *zpg* D59A mutant testes (Fig 6L,6M, 6V and 6W). Testes from *zpg* D21A, *zpg* D21N, or N-terminal delta2-5 truncation mutants all showed a rudimentary testis phenotype that resembled that of *zpg* null flies, no sperm bundles were observed and the flies were sterile (Fig 7H and 7I for wt, 7L–7Q, 7V and 7W, Table 1).

To analyze germ cell differentiation in channel pore mutants they were stained for the germ cell markers Vasa and Boule. In wildtype controls Vasa staining is prominent in the anterior third of the testis (Fig 5H), whereas Boule (Fig 5I) labels the germline in later-stages and is enriched towards the posterior end of the testis where it highlights the parallel, highly organized, arrangement of spermatid tails. In *zpg* null mutants the number of Vasa positive germ

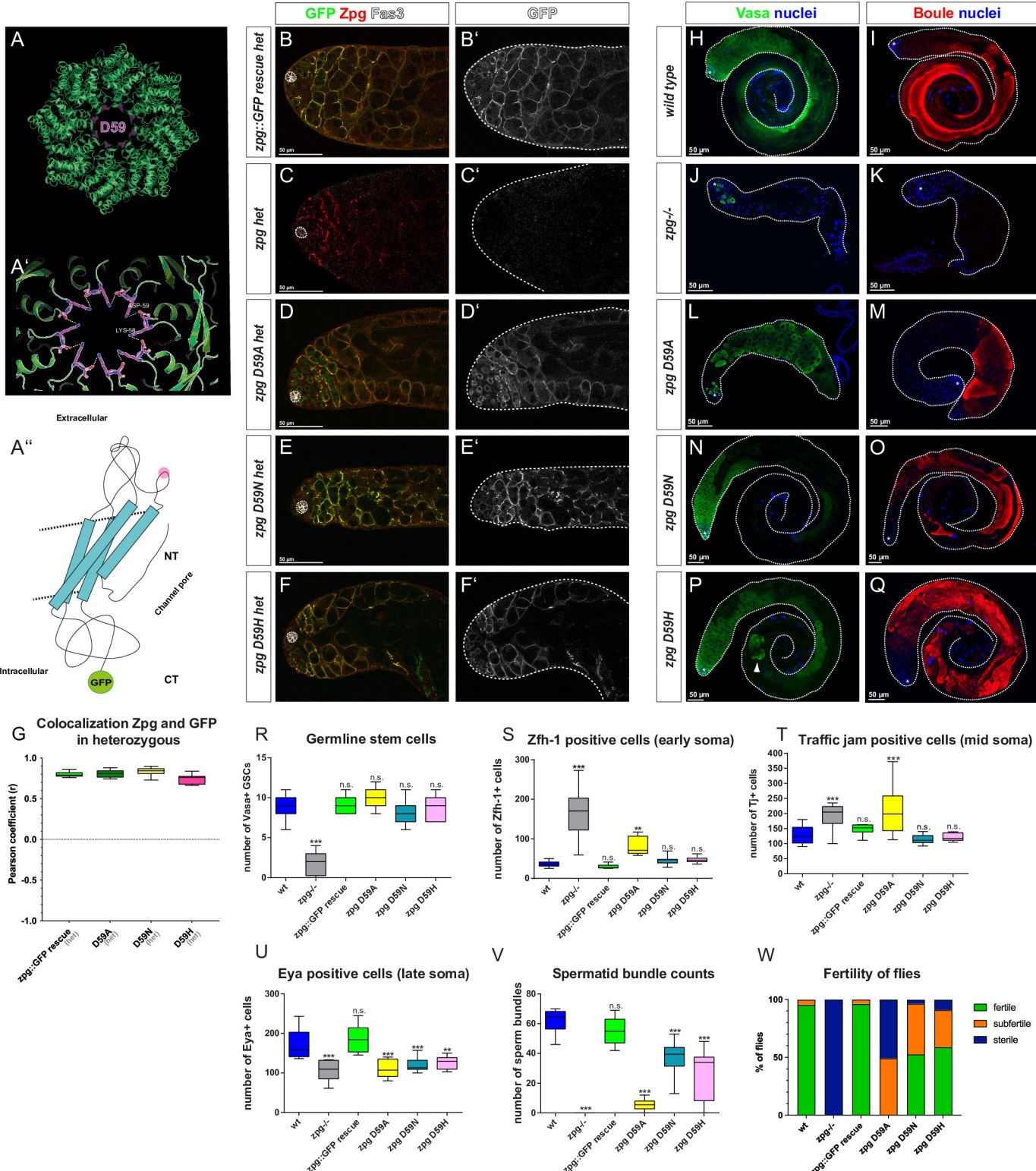

**Fig 6. Mutating the residue D59 of Zpg in the channel pore results in late-stage defects in germ cell differentiation.** (A) Homology model of Zpg showing the position of Aspartate 59 (D59) at the narrowest part of the channel pore, constricting its diameter. (A') D59 is predicted to form a hydrogen bond with lysine (K58) of the neighboring Zpg subunit. (A") Simplified model highlighting the location of D59 in the first extracellular loop of Zpg, facing inside the pore. (B-F) Colocalization of wildtype endogenous Zpg and GFP-tagged, mutated Zpg. Flies heterozygous for a null allele of zpg but also containing one copy of the wildtype genomic *zpg* rescue construct (*zpg* GFP::GR; B-B'), no rescue construct (C-C'), one copy of the *zpg* D59A mutant rescue construct (D-D'), one copy of the *zpg*

D59N mutant rescue construct(E-E'), and one copy of the *zpg* D59H mutant rescue construct (F-F'). The GFP-tagged D59 mutants show strong colocalization with the endogenous Zpg at the membrane. This high degree of colocalization is also revealed by the quantification of the Pearson coefficient between the GFP and Zpg antibody staining (G). (H-Q) Staining for the mitotic germ cell marker Vasa and the late-stage germ cell marker Boule. In the wildtype, Vasa staining is mostly concentrated in the apical part of the testis (H), whereas Boule marks meiotic cysts and long, parallel bundles of spermatids (I). In *zpg* null mutant testes little Vasa (J) and no Boule (K) signal is detected. Early germ cell differentiation defects are detected in *zpg* D59A (L, M), but not in *zpg* D59N (N, O) or *zpg* D59H (P,Q) mutants. Impaired entry to meiosis is seen in testes of *zpg* D59A mutants, since the Vasa signal (L) takes up the entire testis and the Boule signal mostly labels late-stage GC cysts, with very few spermatids (M). Weaker phenotypes are seen in testes of *zpg* D59N and *zpg* D59H mutants, with wildtype Vasa staining (N, P, respectively). However, abnormal cysts are occasionally seen (for example see P, arrowhead). Although Boule staining is rescued in *zpg* D59N and *zpg* D59H mutants they show some disorganization of sperm bundles (O, Q). Quantification of (R) germline stem cells (GSCs), (S) Zfh-1-positive cells, (T) Tj-positive cells, (U) Eya-positive cells, (V) spermatid bundles, and (W) fertility reveals late germ cell differentiation defects in all mutants, with the strongest phenotype seen in *zpg* D59A. Hubs are either encircled or indicated by asterisks. Scale bars represent 50 μm, as indicated above them. n>30 single crosses per genotype for fertility tests. p-values are for difference from wildtype and indicated by asterisks with *p<0.05, **p<0.01, ***p<0.001.

cells (Fig 5J) is low, few early-stage cysts are seen, and no Boule staining was detected due to the early arrest of germ cell differentiation (Fig 5K).

**Mutations in the D50 residue.** In both *zpg* D50A and *zpg* D50R mutants Vasa staining was irregular, filling the entire testis, and multiple large mitotic cysts were seen (Fig 5L and 5N) even in the posterior testis, where mitotic germ cells are not usually found. Boule staining (Fig 5M and 5O) in both *zpg* D50A and *zpg* D50R mutant testes was reduced and spermatid bundles did not appear to be as organized and parallel as in wildtype controls. The *zpg* D50K germline phenotype was similar to that seen in *zpg* null testes, though more early cysts were found compared to *zpg* null testes (Fig 5P and 5Q compared to *zpg* null in 5J, 5K). While both *zpg* D50A and *zpg* D50R had wildtype numbers of GSC, *zpg* D50K flies were similar to *zpg* null mutants in having only few GSCs (average 3.3, 9.0, 9.3 GSCs compared to 9.0 in wt, n = 9, 8, 10 in D50K, D50A and D50R mutants, respectively). It should be noted that the D50A phenotype was quite variable as indicated by the different phenotypes shown in the representative testis shown in Fig 5L, which contains mostly pre-meiotic germ cells, and the testis shown in Fig 5M which contains many elongating spermatids.

**Mutations in the D59 residue.** The early germline phenotype of *zpg* D59N (Fig 6N) and *zpg* D59H (Fig 6P) mutant testes was, in general, wildtype, though on occasion unusual, posterior cysts were seen in *zpg* D59H mutant testes. Late germline stages in D59N and *zpg* D59H mutants also appeared, for the most part, wildtype (Fig 6O and 6Q) with Boule labelling large meiotic cysts and spermatids exhibiting their characteristic ordered arrangement. In contrast *zpg* D59A mutants did not appear wildtype and exhibited Vasa positive cysts abnormally located throughout the testes with only a few late spermatid stage cysts (Fig 6L and 6M). Nonetheless, GSC maintenance was not impacted by any of the D59 mutants, *zpg* D59N, *zpg* D59H, or *zpg* D59A (average 8.2, 8.7, 10.1 GSCs compared to 9.0 in wt, n = 15, 9, 15 in D59N, D59H and D59A mutants, respectively).

**Mutations in the D21 residue and N-terminal truncation.** Testes from flies expressing mutations in the D21 residue in Zpg, *zpg* D21N and *zpg* D21A, as well as the N-terminal truncation mutants, *zpg* delta 2–5, appeared rudimentary with few Vasa positive germ cells (Fig 7L, 7N and 7P) and no detectable Boule staining (Fig 7M, 7O and 7Q), consistent with a strong loss of function or null *zpg* phenotypes. Quantification of GSCs showed reduced numbers, similar to *zpg* null mutants, in *zpg* D21N and *zpg* D21A mutants, as well as the N-terminal truncation mutants, *zpg* delta 2–5 (Fig 7R, average 2.4, 4.0, 2.1 GSCs, n = 8, 20, 18 in D21N, D21A and delta 2–5 mutants, respectively).

Overall, as summarized in Table 1, we observe a complex spectrum of germline phenotypes in our channel pore mutants, some behave like *zpg* null mutants while others behave as nearly wildtype with a range of intermediate phenotypes in between. This is in line with our initial expectation and goal of generating mutations that would modulate the passage of various cargoes and therefore impinge on different aspects of spermatogenesis.

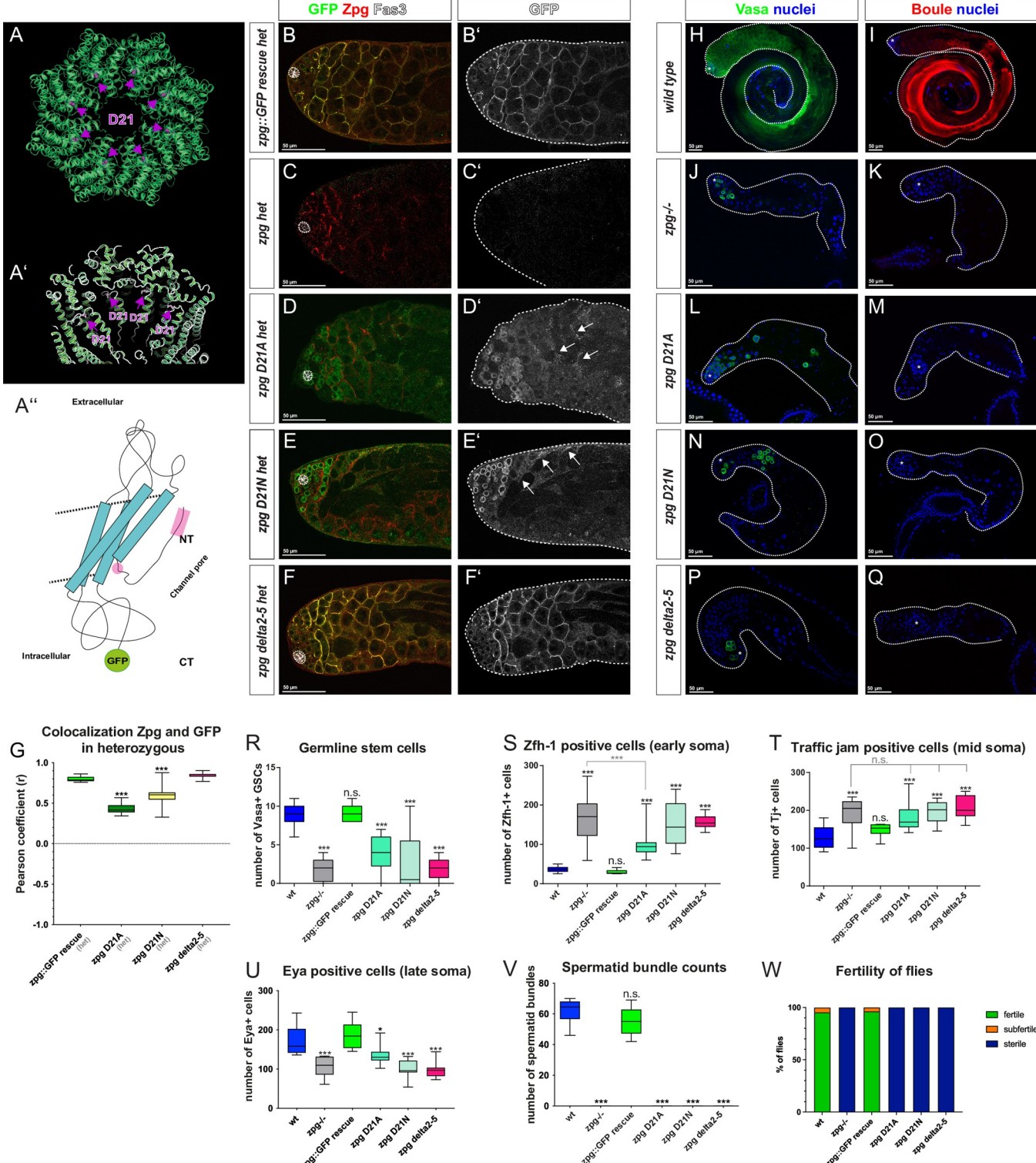

**Fig 7. Mutations in the channel-gating N-terminus of Zpg lead to loss of function.** (A) Homology model of Zpg showing the position of Aspartate 21 (D21) residue within the Zpg channel reveals a localization within the channel pore. (A') The side chains of D21 are not in proximity to any other amino acids. This makes a direct interaction with the cargo likely. (A") The introduced N-terminal mutations are highlighted in pink in a schematic view of the Zpg protein structure. In the deletion mutant *zpg* delta2-5, the highly flexible N-terminal domain was shortened, while D21 sits at the hinge between the N-terminal chain and the first transmembrane domain. Due to the limited number of germ cells in the N-terminal mutants, which makes it hard to assess subcellular localization, the localization of Zpg (GFP-tagged; green in B-F, single channels in B'-F') was analyzed in testes of flies harboring one copy of the respective

mutation and once copy of endogenous Zpg (red in B-F). In the control (*zpg*::GFP GR, B-B'), the mutated and endogenous Zpg colocalize at the plasma membrane. In non-rescued flies heterozygous for *zpg* (C-C'), Zpg localizes to the membrane and no GFP-tagged construct is expressed. Testes of *zpg* D21A (D-D') and *zpg* D21N (E'E') mutants display weak membrane localization of the mutated proteins (see arrows), while the majority of the signal is cytoplasmic. In contrast, strong colocalization is found in *zpg* delta2-5 mutants (F-F'). (G) Measurement of the Pearson coefficient for colocalization of Zpg and GFP in testes of heterozygotes with one copy of endogenous and one copy of mutated Zpg. Strong colocalization in *zpg* delta2-5 mutants, similar to the *zpg*::GFP GR control, is observed, but only weak colocalization in *zpg* D21A and *zpg* D21N mutants. (H-Q) Staining for the mitotic germ cell marker Vasa and the late-stage germ cell marker Boule. In the wildtype, Vasa staining is mostly concentrated in the apical part of the testis (H), whereas Boule marks meiotic cysts and long, parallel bundles of spermatids (I). In *zpg* null mutant testes little Vasa (J) and no Boule (K) signal is detected. Testes of *zpg* D21A (L-M), *zpg* D21N (N-O), and *zpg* delta2-5 (P-Q) mutants exhibit a *zpg* null mutant-like phenotype. Quantification of (R) germline stem cells (GSCs), (S) Zfh-1-positive cells, (T) Tj-positive cells, (U) Eya-positive cells, (V) spermatid bundles, and (W) fertility reveals null mutant-like phenotypes in all three N-terminal mutants, the exception being partial rescue in the number of Zfh-1 positive cells in *zpg* D21A mutants. Hubs are either encircled or indicated by asterisks. Scale bars represent 50 μm, as indicated above them. n>30 single crosses per genotype for fertility tests. Unless otherwise indicated p-values are for difference from wildtype and indicated by asterisks with *p<0.05, **p<0.01, ***p<0.001.

## Diverse somatic phenotypes observed in Channel pore mutations in Zpg

To explore links between specific disruptions of cargo transport through the gap junctions and somatic differentiation, the expression of various somatic markers was studied in channel pore mutants:

**Mutations in the D50 residue.** In testes of *zpg* D50A and *zpg* D50R mutants, early somatic cell differentiation was mildly dysregulated, with a higher number of cells positive for the early marker Zfh-1 at the apical tip (Figs 5S, S6L and S6O; means of 84 and 50, n = 10 and 8 for the D50A and D50R mutants, respectively) but normal staining patterns and cell counts for Tj (Figs 5T, S6M and S6P; mean 149 and 152, n = 10 and 12 for D50A and D50R mutants, respectively). The number of cells positive for the late marker Eya however, was significantly

**Table 1. Summary of the mutants generated in this study.**

| Mutant | Target | Localization | Rescue? | Additional remarks |
|---|---|---|---|---|
| *zpg*::*GFP* rescue (genomic rescue) | No mutation (rescue construct) | at membrane | full rescue | Full length rescue construct (proof of principle) |
| *zpg* deltaCT::GFP | C-terminus | cytoplasmic | none detected | Deletion of the entire C-term |
| *zpg* Y352F | Phosphorylation site | at membrane | full rescue | |
| *zpg* phospho dead | Both phosphorylation sites (Y352 and S356) | at membrane | full rescue | |
| *zpg* C6S | Docking to another Inx | cytoplasmic | none detected | |
| *zpg* C236S | Docking to another Inx to form GJ | cytoplasmic | none detected | |
| *zpg* C145S | Docking to another Inx to form GJ | cytoplasmic | none detected | |
| *zpg* C6S::GFP | Docking to another Inx to form GJ | cytoplasmic | none detected | |
| *zpg* C26S::GFP | Docking to another Inx to form GJ | cytoplasmic | none detected | |
| *zpg* D50A::GFP | Channel pore | at membrane | partial rescue | Germ Cell differentiation defects during meiosis |
| *zpg* D50R::GFP | Channel pore | at membrane | partial rescue | Germ Cell differentiation defects during meiosis |
| *zpg* D50K::GFP | Channel pore | at membrane | minimal rescue | |
| *zpg* D59A::GFP | Channel pore | at membrane | partial rescue | Germ Cell differentiation defects during meiosis |
| *zpg* D59H::GFP | Channel pore | at membrane | partial rescue | Germ Cell differentiation defects during individualization |
| *zpg* D59N::GFP | Channel pore | at membrane | partial rescue | Germ Cell differentiation defects during individualization |
| *zpg* D21A::GFP | Channel pore | Both membrane & cytoplasmic | none detected | D21 predicted to directly interact with cargo |
| *zpg* D21N::GFP | Channel pore | Both membrane & cytoplasmic | none detected | D21 predicted to directly interact with cargo |
| *zpg* delta2-5::GFP | Channel pore | at membrane | none detected | N-terminal deletion (amino acid 2–5) |

reduced (Figs 5U, S6N and S6Q; mean 111 and 132, n = 8 and 7, for D50A and D50R mutants, respectively). Stronger somatic phenotypes, comparable to a null *zpg* mutant, were observed for the D50K mutation, both Zfh-1 (Figs 5S and S6R; mean 194, n = 10) and Tj (Figs 5T and S6S, mean 200, n = 8) counts were elevated, whereas the number of Eya positive cells was reduced compared to controls (Figs 5U and S6T; mean 98, n = 8).

**Mutations in the D59 residue.** A similar mix of early and late phenotypes was seen with different mutations targeting the D59 residue in the channel pore. Both the *zpg* D59N and *zpg* D59H mutant fly lines, exhibit relatively normal early somatic differentiation, as judged using the somatic markers Zfh-1 (Figs 6S, S7O, S7RR; means of 45 and 46, n = 11 for both *zpg* D59N and *zpg* D59H), and Tj (Figs 6T, and S7P, S7S; mean 114 and 120 for n = 10 and 8 for *zpg* D59N and *zpg* D59H, respectively). In comparison, later somatic development, analyzed using the late marker Eya, was disrupted, though not as severely as in null *zpg* mutants (Figs 6U, and S7Q, S7T; mean 120.8 and 125.5, n = 10 and 11, for *zpg* D59N and *zpg* D59H, respectively). Stronger phenotypes closer to *zpg* null mutants were obtained in *zpg* D59A mutant testes with a higher number of both Zfh-1 positive cells (Figs 6S, and S7L; mean 81, n = 20) and Tj positive cells (Figs 6T and S7M; mean 208, n = 24), and a lower number of Eya positive cells (Figs 6U and S7N; mean 111, n = 9) compared to controls.

**Mutations in the D21 residue and N-terminal truncation.** Mutations in the D21 channel pore residues, as well as the N-terminal delta2-5 truncation, gave rise to strong *zpg* null-like somatic phenotypes. Specifically, there were higher numbers of both Zfh-1 positive cells (Figs 7S, and S8L, S8O, S8R; mean 150, 99, and 156, n = 8, 14, and 10 for D21N, D21A, and the delta2-5 mutations, respectively) and Tj positive cells (Figs 7T, and S8M, S8P, S8S, mean 197, 180, and 207, n = 10, 13, and 10 for D21N, D21A, and the delta2-5 mutations, respectively), and a corresponding decrease in the number of Eya positive cells (Figs 7U, and S8N, S8Q, S8T; mean 136, 100, and 96, n = 7, 13, and 13 for D21N, D21A, and the delta2-5 mutations, respectively). Overall, as summarized in Table 1, these results show that mutations that selectively disrupt signals that move through the gap junction channel pore from the germline to the soma produce distinct somatic phenotypes.

## Intermediate alleles of channel pore mutations identify a function for gap junction in sperm individualization

A number of the channel pore mutants reached the late stages of spermatogenesis yet exhibited noticeably reduced fertility. These included the largely wildtype appearing *zpg* D50A, *zpg* D50R, and *zpg* D59N mutants, as well as the more severe *zpg* D59A mutant (Figs 5V, 5W, and 6V, 6W).

Since these phenotypes were consistent with late arising spermatogenesis defects, we analyzed the spermatid stages in greater detail. Phase contrast microscopy was used to analyze "onion stage" spermatids, and in particular nuclear phenotypes and nebenkern numbers (Fig 8A–8F). With the exception of *zpg* D59N mutant testes (Fig 8F) which appeared wildtype, *zpg* D50A, *zpg* D50R, *zpg* D59A, and *zpg* D59H mutants all exhibited multinucleation defects and abnormal nucleus to nebenkern ratios in some of their spermatids. To study individualization, we looked at the sperm actin caps in intermediate strength channel pore mutants (Fig 8G–8L). While in the wildtype there is tight association between the actin caps in freshly dissected sperm (labelled with phalloidin) and the nuclei (labelled with TO-PRO-3) in all intermediate strength channel pore mutants, we observed abnormal actin caps that had only loose association between actin and nucleus. To quantify this phenotype, we measured the angle between the actin filaments and the nuclei to determine the degree of organization of the spermatid bundles (Fig 8M, n = 45 for all genotypes except for *zpg* D59A (n = 20)). In wildtype controls

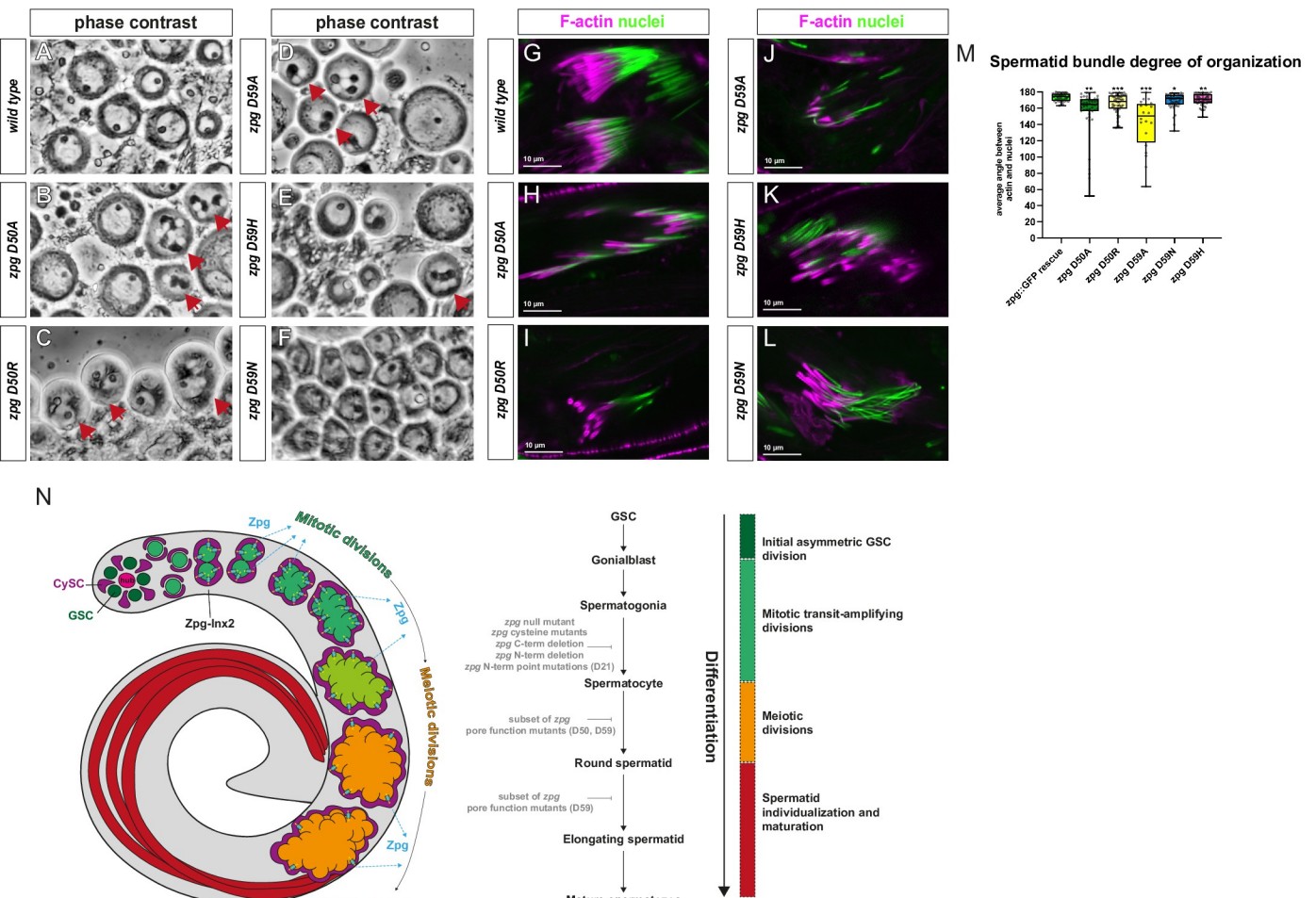

**Fig 8. Defects in the spermatid stage are detected in a subset of *zpg* channel pore mutants.** (A-F) Phase contrast imaging of "onion stage" spermatids in live testes in wt and in the subset of *zpg* mutants that show intermediate germline differentiation defects. In the wildtype (A), spermatids appear as round cells with large nuclei with a small nebenkern (black dot inside nuclei). Major defects such as multinucleation and abnormal nucleus-to-nebenkern ratios are seen in spermatids of *zpg* mutants rescued with the D50A (B), D50R (C), D59A (D), and D59H (E) mutant *zpg* genomic rescue constructs, as indicated by red arrows. Spermatids of *zpg* D59N mutants (F) do not show nuclear defects. (H-L) Spermatid individualization complexes (ICs) in testes of freshly eclosed flies were stained with rhodamine phalloidin (labels actin cones, magenta) and TO-PRO-3 (labels nuclei, green). In the wildtype (H), the actin cones are tightly associated with the elongated nuclei of 64 developing spermatids. In testes of *zpg* D50A (H), *zpg* D50R (I), *zpg* D59A (J), *zpg* D59H (K) and *zpg* D59N (L) mutants, the association of actin can be disrupted and the overall IC structure appears disorganized. Scale bars indicate 10 μm. (M) Quantification of the angle of association between the actin cones and the nuclei in ICs. The wildtype has a 180° angle between the actin and the nuclei due to the linear organization of the IC. In all analyzed mutants, but in particular in *zpg* D50A and *zpg* D59A, the angle is smaller, indicating disorganization. n = 45 for all mutants except *zpg* D59A (n = 25 due to lower abundance of spermatid bundles). (N) Simplified model summarizing of the function of Zpg during spermatogenesis. Zpg is required for all major developmental transitions in fly spermatogenesis. Schematic depicts the process of spermatogenesis starting at stem cell the stem cell niche at the apical tip of the testis and ending at differentiated mature sperm (GSCs, dark green; CySCs and cyst cells, magenta; hub; pink). The gap junction (cyan) consisting of Zpg and Inx2 is found at the soma-germline interface and allows bi-directional passage of cargo (yellow arrows) between soma and germline. Soma-germline communication is required for the first, mitotic steps of germ cell division, as *zpg* null mutants fail to enter the transit-amplifying stages. Zpg-mediated soma-germline communication is also required in later stages of germ cell differentiation, since germ cells in hypomorphic *zpg* mutants generated in this study failed to enter and properly execute meiosis and/or spermatid individualization. The different stages of spermatogenesis are indicated by a colour code (Green: early stages, yellow: mid stages, red: late stages).

there is typically a 180-degree angle between the actin filaments and the nuclei, indicating a linear alignment. In comparison, all the intermediate channel pore mutants, and in particular the *zpg* D59A and the *zpg* D50A mutants, exhibited a smaller angle indicating various degrees of disorganization. Taken together this data shows that mutations that selectively disrupt signals that move through the gap junction channel pore from the germline to the soma can disrupt specific signals in late-stage spermatogenesis that are required to organize actin and facilitate sperm individualization.

## Discussion

In this study, we took advantage of the powerful genetic tools of *Drosophila* and the relatively small size of the genomic DNA region required to fully rescue *zpg* to perform what is, to our knowledge, the most comprehensive structure function analysis of a gap junction protein in an *in vivo* context. Taking advantage of the large body of biochemical and functional studies of gap junction proteins, we were able to identify and disrupt specific key regulatory and functional sites within *zpg* and define their role in soma-germline communication during spermatogenesis. Spermatogenesis is particularly suitable for this type of structure function, as encapsulation of the germ cells by somatic cells completely isolates the germline from the environment [9,19]. This means that the germline is fully reliant for its survival on gap junction-mediated transport of external cues required for cell differentiation and/or nutrients and metabolites. Our analysis provides several key insights into how Innexins work as well as into how spermatogenesis is regulated. First, we show that the C-terminal of *zpg* is required for its transport and/or stabilization at the plasma membrane but that C-terminal phosphorylation sites are not essential for function. Second, we provide strong *in vivo* evidence that Innexin function requires gap junction formation similar to connexins but not to the more closely related pannexins, which function as hemichannels. Third we observe that mutations that are designed to target the channel pore, and selectively alter the passage of regulatory molecules through the gap junctions, produce a range of phenotypes spanning from early to late stages of spermatogenesis. This identifies an important role for N-terminal mediated cargo selectively which suggests that different gap-junction mediated cargoes regulate specific stages of sperm differentiation. More broadly, these results identify roles for gap junction-mediated soma-germline communication in multiple stages of spermatogenesis, consistent with the continuous role for gap junctions throughout sperm development.

A general obstacle for mechanistic studies of gap junctions is the difficulty in identifying the specific cargoes that pass through the channel pore This task is made even more difficult by noting that, multiple gap-junction cargoes can be used simultaneously to control a specific biological outcome. As a partial workaround to this issue we used structural and biochemical insights to selectively alter the channel pore with the hope of identifying mutations that selectively inhibit specific cargoes. This would allow us to study the role of gap junctions in diverse processes without the need to identify the entire set of cargoes that are required in every such process. We propose that some of the mutations we tested indeed fulfill the criteria for such alterations. For example, the *zpg* D50A, *zpg* D50R, and *zpg* D59N go through a fairly normal early spermatogenesis, in terms of both somatic and germline development, but exhibit specific defects when they reach the key late developmental process of sperm individualization. The D50 and D59 residues are located in the narrowest part of the channel pore and changes in these residues would be predicted to alter the ability of different cargos to pass through the channel. In comparison, two, more drastic mutations, *zpg* D59A and *zpg* D59H, exhibit stronger phenotypes that are a mixture of early and late spermatogenesis defects. These results are in line with a model in which specific disruptions in the passage of different types of cargoes impinges on spermatogenesis in distinct ways and illustrates how unique cargoes are required to modulate early versus later stages of sperm development.

Analysis of the phenotype in mutants lacking *zpg* expression [51–53] has shown that in *zpg* mutants, the early stages of germ cell differentiation were blocked. This early and severe phenotype of *zpg* null mutants made the analysis of the function of Zpg in later stages of spermatogenesis impossible. In a number of the hypomorphic mutants we generated in the present study, germ cell differentiation defects were more complex. In particular our data suggests defects in the ability of some channel pore mutants to enter and properly execute meiosis.

Mutants such as *zpg* D50A and *zpg* D50R, produce a version of the Zpg protein that localized to the plasma membrane, and exhibited early somatic and germline phenotypes that are, based on our analysis, wildtype. However, they develop defects during the germline's transition from mitosis to meiosis. There are examples in the literature of gap junction-mediated signaling initiating or regulating meiosis, such as oocyte maturation in rodents [69,70]. Oocyte maturation is inhibited by cAMP through a kinase cascade and high levels cAMP are maintained using inhibition of the activity of cAMP phosphodiesterase through gap junction derived cGMP [71–73]. Upon exposure to LH (luteinizing hormone), cGMP diffuses out of the oocyte through gap junctions, causing lower levels of cAMP and triggering the reentry to the meiotic cell cycle [74]. Similarly, gap junction mediated signals seem to be essential for the proper assembly and/or maintenance of actin caps during late spermatogenesis, an intricate process that requires close coordination between the soma and the germline [75]. In sum, the identification of new, late acting alleles of Zpg provides an opportunity to identify and study novel roles and mechanisms of action of soma germline communication.

The intracellular C-terminus domain is a an important binding site for interacting proteins and plays a role in channel gating [34,35]. The functional diversity of the C-termini of innexins [20] and connexins [33] is reflected by the observation that these domains vary greatly in length and constitute the least conserved domain among innexins and connexins. A truncated version of Zpg lacking the C-terminal domain, deltaCT::GFP, failed to localize to the membrane of germ cells and became trapped intracellularly, resulting in complete loss of function. This implies that the C-terminus might be required for either delivering or stabilizing Innexins at the membrane, potentially through facilitating interactions with binding partners. Though it is possible that protein folding is impaired by the truncation, truncating the C-terminal domain in other gap junction does not affect the ability of the protein to fold properly. For example, in mouse cardiomyocytes, truncation of the C-terminus of the Connexin CX43 did not prevent it from either localizing to the membrane or transmitting current between cells [76]. Intriguingly though, this truncation did exhibit phenotypes consistent with changes in CX43 stability and turnover, namely, fewer, but larger gap junction plaques were observed. Complicating this picture is the observation that different tissues react differently to C-terminal truncations, for example, in contrast to cardiomyocytes in the mouse neocortex or epidermis, C-terminal truncations in CX43 resulted in loss of function [77,63]. Analyzing the function of the C-terminus of Zpg in additional tissues as well as the identification of potential interaction partners remains an important subject for future investigations.

Gap junction proteins couple to each other via disulfide bridges formed between cysteine residues in their extracellular domains [22,23]. Vertebrate pannexins, which are closely related to invertebrate innexins, only form hemichannels, thereby allowing the flow of small molecules between cytoplasm and extracellular space [24]. Several human connexins can form hemichannels as well, and though there has been speculation innexins can also function as hemichannels this has not been directly tested [25,78]. Our data shows that Zpg is very sensitive to disruptions in the Cysteine-residue coupling that mediates gap junction formation as mutating even a single Cysteine residue severely disrupt protein function. This strongly supports the assertion that Zpg functions during spermatogenesis predominantly by forming gap junctions between the soma and the germline. Furthermore, this observation allowed us to ask how mutations that affected gap junction gating modulated soma-germline communication. Based on our predicted 3D structure, and in line with studies using *C. elegans* INX-6 [55], the short N-terminal domain and the first part of the Extracelullar Domain 1 of Zpg are located inside the channel pore. We focused on three residues, D21, D50 and D59, located inside the pore, that are predicted to either directly interact with positively charged cargo passing through the channel (D21) or to regulate pore permeability by forming hydrogen bonds with other amino

acids at the narrowest, size-limiting part of the pore (D50, D59). All channel pore mutants were able, to varying extents, to localize to and be retained at the membrane, suggesting that mutating residues within the channel pore does not substantially interfere with protein folding and trafficking to the membrane. It has been shown for several human connexins that the passage of cargoes through the channel is regulated by amino acids found within the pore, as these residues form a network of stabilizing hydrogen bonds [41]. The interactions that occur within the channel pore are complex, and it is hard to predict the behavior of the mutants. For example, it was surprising that while K and R are not greatly different in size or charge, the germ cell differentiation phenotype of *zpg* D50K mutants was more severe than the one in D50R mutants. This highlights the usefulness of the kind of trial-and-error approach we were able to adopt for Zpg structure/function analysis, where we could study multiple mutations due to the relative ease of generating mutant rescue constructs.

Another of the N-terminal mutants we generated, a deletion of the first four amino acids, excluding the methionine, provided additional mechanistic insight into Zpg function. In other gap junction proteins the N-terminal helix is a highly flexible structure that is actively involved in opening and closing the channel [67,39,55,56]. If this was the case for Zpg than a deletion of the first part of the helix would result in constitutive opening of the channel. Alternatively, the deletion of the four amino acids might critically interfere with the gating mechanism and disturb the structure of the channel, thereby impairing the gating mechanism and blocking the passage of cargo through the channel. For the *zpg* delta2-5 deletion mutant, we observed a loss of function phenotype supporting the latter model. This result is consistent with structural studies showing that the N-terminus forms a funnel-like pore structure of a specific size and electrostatic charge, that is a central regulator of channel conductance and cargo selectivity [41,55,56,79].

Based on the data we obtained as well as previous studies of Zpg function we propose the following model (Fig 8N). Soma-germline communication, mediated by gap junction channels consisting of Zpg on the germ cell membrane and Inx2 on the somatic cell membrane, is required at multiple stages of germ cell differentiation. Once the developing early germ cells are encapsulated by somatic cells and are closed off from their environment [9,19], they fully rely on signals passed through the gap junction channel, which regulate their differentiation, nourish them, and are required for their survival. Only when soma-germline communication is intact, can the germ cells proceed to differentiate and divide, first entering the transit amplifying stages of mitotic divisions. As the cysts grow larger, Zpg is still required for further divisions. In this study we, for the first time, demonstrate the function of soma-germline communication at the transition to meiosis, as hypomorphic *zpg* mutants show defects in the initiation of meiotic divisions, resulting in reduced fertility. Finally, we identify a role for gap junction-mediated soma-germline communication in sperm individualization. This implies that Zpg-mediated soma-germline communication plays a crucial role not only in initiating the first round of divisions, but also in triggering the switch to meiotic divisions and in the normal progress through sperm individualization, which makes Zpg a central regulator of developmental transitions at multiple points during spermatogenesis. In this regard our study may have potential implications to an important question in the field of mammalian spermatogenesis. Specifically, it is not currently known how Sertoli cells, which encapsulate the germ cells through all stages of mammalian spermatogenesis can have essential regulatory roles in all the different steps of germ cell differentiation. It is remarkable that at each developmental transition of mammalian spermatogenesis, from spermatogonia to spermatocytes and then to spermatid, Sertoli cells have a major and clearly defined contribution [80]. Since it is known that gap junction proteins, most notably the Sertoli cell specific connexin 43, are required for multiple stages of spermatogenesis [48], we can envision a similar mechanism to that we

observe in the fly testes. In particular, it may that distinct gap-junction mediated signals, each acting at a specific stage of spermatogenesis, regulate each sequential step in mammalian spermatogenesis.

## Materials and methods

### Fly lines and husbandry

The following lines were used: $zpg^{z-2533}$, st/TM3, Sb; $zpg^{z-5352}$/TM6B; multibalancer lines as well as the fly lines generated in this study (see below). Flies were kept on a standard diet. Crosses were set up at room temperature, were kept at 25˚C and transferred to fresh food every other day. To generate *zpg* null mutant flies, $zpg^{z-2533}$/TM3, Sb, YFP flies were crossed to $zpg^{z-5352}$/TM3, Sb, YFP flies. For the wild type control, $zpg^{z-2533}$/TM3, Sb, YFP flies were crossed to $w^{1118}$.

All newly generated mutant genomic rescue fly lines (zpg GR) were crossed into the $zpg^{z-5352}$ background, generating the following genotype: +/y; zpg $GR^{Mut}$/CyO; $zpg^{z-5352}$/TM6B. Males of these lines were crossed to virgins of $zpg^{z-2533}$/TM3, Sb, YFP. The offspring was selected for markers and the following genotype was chosen for analysis: +/y; zpg $GR^{Mut}$/+; $zpg^{z-2533}$/$zpg^{z-5352}$. For the analysis of heterozygous flies that still have one copy of the endogenous *zpg*, but also one copy of the respective mutant rescue construct, the following genotype was chosen: +/y; zpg $GR^{Mut}$/+; $zpg^{z-2533}$/TM6B.

### Generation of fly lines

All fly lines generated in this study were based on the rescue construct described in Smendziuk et al., 2015 [53]. Briefly, a 6.15kb genomic fragment including the *zpg* locus [51] and an additional 1.5kb both upstream and downstream were cloned into the pAttB vector. This was used as template for site directed mutagenesis to introduce deletions and point mutations to the *zpg* gene. Many of the generated constructs (see Table 1) contain an additional a GFP cassette preceded by a short linker sequence (LAAA).

For site directed mutagenesis, the Q5 Mutagenesis Kit (NEB) and the QuikChange XL Mutagenesis Kit (Agilent) were used according to the manufacturers' instructions. Plasmids were purified using the PureLink HiPure Plasmid Midiprep Kit (ThermoFisher). Constructs were verified by Sanger sequencing prior to injection (Bestgene) and inserted into the fly genome via the attP40 integration site on the second chromosome. Transformants were selected by eye colour and crossed to flies of a multi-balancer line to generate stable, balanced stocks. A list of generated mutants can be found in Table 1.

### Homology modeling

Sequences of *Drosophila* and *C. elegans* innexins were aligned using CLC Workbench Software and Protein BLAST. For protein homology modeling, the protein structure of Zpg (Uniprot Q9VRX6) was modeled onto the previously published structure of *C. elegans* INX6 ([55]; PDB entry 5H1Q) using SWISS-MODEL. The 3D protein structure model was assessed and figures were generated using Pymol software.

### Fertility assays

For assessing fertility of male flies, three $w^{1118}$ virgin females, 4–5 days post eclosion (dpe) were mated with a 1dpe male for two days. Afterwards, mated flies were transferred to fresh food for 24 h and kept at 25˚C. The number of offspring from this 24 h period was counted

**Table 2. Antibodies used in this study.**

| Antibody/Dye | Source | Species | Dilution |
|---|---|---|---|
| Zpg | Smendziuk et al., 2015 [53] | rabbit | 1:20000 |
| GFP | Abcam, ab13970 | chicken | 1:1000 |
| Vasa | R. Lehmann | rabbit | 1:5000 |
| Boule | S. Wasserman | rabbit | 1:1000 |
| Zfh-1 | J. Skeath | guinea pig | 1:500–1:1000 |
| Traffic jam (tj) | D. Godt | guinea pig | 1:5000 |
| Eya | DSHB, 10H6 | mouse | 1:250–1:500 |
| Fasciclin III (Fas3) | DSHB, 7G10 | mouse | 1:500 |
| DN Cadherin | DSHB, DN Ex #8 | mouse | 1:50 |
| TO-PRO-3 | Thermo Fisher | | 1:500 |
| Rhodamine Phalloidin | Abcam, ab235138 | | 1:100 |

DSHB = Developmental Studies Hybridoma Bank

and categorized as follows: Fertile: more than 30 offspring/day; Subfertile: less than 30 offspring/day; Infertile: 0 offspring/day.

## Immunostaining and confocal imaging

Dissection of testes of adult male *Drosophila* was performed as previously described [53]. Briefly, <1dpe males were vivisected in 1x PBS pH 7.4 and fixed in 4% PFA for 20 min. Samples were washed and antibody solutions were prepared in 1x PBS supplemented with 0.5% BSA and 0.3% Triton X-100. Samples were incubated with primary antibodies overnight at 4˚C. See Table 2 for a list of primary antibodies used in this study. Secondary antibodies (Jackson Immuno; coupled to A488, Cy3, or A647/Cy5) were diluted 1:500 and incubated in the dark for 2 hours at room temperature or overnight at 4˚C.

Samples were mounted in Vectashield H-1200 with DAPI or in Vectashield H-1000 (from Vector Laboratories). For nuclear labelling, either Vectashield H-1200 was used or the nuclear dye TO-PRO-3 (1:500, ThermoFisher) was incubated together with the secondary antibodies.

For labelling of individualization complexes, testes were dissected, fixed in PFA, washed and then incubated in a solution containing Rhodamine Phalloidin (1:100, abcam) and TO-PRO-3 (1:500, ThermoFisher) for 2h. After a 15 min wash in PBS, samples were mounted in Vectashield H-1000 and immediately subjected to imaging.

Confocal images were taken using an Olympus FV1000 inverted confocal microscope with an UplanSApo 20x0.75, an UplanFL N 40×, 1.30NA oil objective, and an UplanSApo 60×, 1.35 NA oil objective. Image analysis was performed in Olympus Fluoview and in FIJI, cell counts were performed in MatLab. Figures were assembled in Adobe Illustrator 2020.

## Image analysis

Cell counting was performed in MatLab using a script optimized for 3D cell counting. Briefly, Z-stacks are combined into a tensor to filter and perform a 3D distance transform and watershed to identify and count cells. Each 2D Z-stack is also counted to see cell density. The script can be found at GitHub: (https://github.com/Tanentzapf-Lab/GapJunction_Spermatogenesis_Pesch). GSCs and Zfh-1-positive cells were counted manually using the Cell Counter plugin in FIJI. For cell counts, GSCs were defined as Vasa-positive single cells contacting the hub. CySCs were defined as Zfh-1-positive cells in direct proximity (<10µm) to the hub. Fluorescent intensity quantifications were performed in FIJI. For determining the degree of

colocalization between the endogenous Zpg and the GFP-tagged mutant Zpg in heterozygous fly testes, the JACoP plugin [81] in FIJI was used to determine the Pearson colocalization coefficient. For quantifying the degree of organization of individualization complexes, the angle between F-actin and nuclei was measured in FIJI using the angle tool. For imaging of spermatid bundles, testes were stained with DAPI or TO-PRO-3 to visualize the needle shaped nuclei of elongated spermatids (S4F Fig). Each of the clusters of 64 syncytial nuclei was counted as one spermatid bundle using the Fiji Cell counter plug-in. GraphPad Prism 9 was used for statistical analysis.

### Imaging of live spermatocytes and spermatids

For live imaging of late-stage germ cells, freshly eclosed male flies were vivisected in 1x PBS pH 7.4 and the testes were transferred into drops of PBS on microscope slides. A coverslip was added and the samples were gently squished. For brightfield imaging, a phone attachment (Smartphone Digiscoping Adapter, Gosky Optics) was mounted onto an Olympus CKX53 microscope and pictures were taken using the 20x objective and 4x zoom.

### Statistical analysis

Mean and standard error of the mean (SEM) are shown. All statistical analysis was performed in GraphPad Prism 9 using one-way ANOVA with multiple comparisons test (Tukey) or using unpaired t-tests with Welch's correction, respectively. P-values indicated are $^*p<0.05$, $^{**}p<0.01$, $^{***}p<0.001$.

### Supporting information

**S1 Fig. Homology modelling of the Zpg/Inx4.** a) Model of a single subunit of *D.melanogaster* Inx4, color coded according to reliability on a scale from dark red (0% Swiss Model score) to dark blue (70% or higher). This shows that the transmembrane region is the most reliable. Of note, a very similar model is obtained through Alphafold (B, C), adding further confidence to the overall fold and general location of the three Asp residues investigated in this study. The positions of Asp21, Asp50, and Asp59 are indicated via purple sticks. (B) Superposition of models for *D.melanogaster* Inx4 derived from homology modeling (cyan) and through Alphafold2 (red), showing excellent agreement in the fold, especially for the transmembrane region. (C) model for one subunit of Inx4 obtained through Alphafold2, color coded according to reliability score (from 40% in red, to 100% in dark blue). The areas with lowest reliability include the N-terminus and C-terminus, and a small extracellular loop (circled). As expected, these areas also show the largest divergence between the two models.
(TIF)

**S2 Fig. Schematics representation of the Zpg protein showing key sites for structure-function analysis.** (A) 2D plot of the topology of Zpg marking residues and domains of interest for our structure-function analysis. The Zpg gap junction is a 4-pass transmembrane protein with intracellular N- and C-termini. While the N-terminus and the first extracellular helix are predicted to face inside the channel pore where they likely regulate channel gating, the C-terminus likely has a channel-independent function and contains two phosphorylation sites for potential phospho signaling. It is known from other innexins that cysteine residues in the extracellular region mediate the coupling of two hemichannels in adjacent cells. Here, Zpg in the germ cell membrane couples to Inx2 in the soma cell membrane. (B-B''') Simplified schematics of the Zpg protein showing the site of GFP-tag insertion for the wildtype (*zpg*::GFP GR rescue flies, B), and C-terminal deletion mutant (*zpg*::GFP deltaCT; B'). Also shown are sites of residues

altered in the phosphorylation mutants (*zpg* Y352F and *zpg* Y352F/S356A; B") and three different mutations of extracellular cysteines (*zpg* C6S, *zpg* C145S, *zpg* C236S; B"').
(TIF)

**S3 Fig. A single copy of *Zpg*::GFP GR results in expression levels that are similar to those seen in *zpg* heterozygous mutant flies.** (A) Fluorescence intensity quantification (n = 12 images measured per genotype; mean fluorescence intensity in arbitrary units) of Zpg antibody staining in testes of wildtype control, flies heterozygous for the *zpg*$^{2533}$ null allele (*zpg*$^{2533}$/+) and zpg mutants rescued with one copy of the genomic rescue construct (*zpg*::GFP GR). Similar levels of expression are observed in testes of *zpg* mutants rescued with *zpg*::GFP GR and *zpg*$^{2533}$ heterozygotes. (B-B') Representative images showing expression levels for wildtype (B) and *zpg* mutants rescued with *zpg*::GFP GR (B'). Hubs are marked by asterisks. Scale bars represent 50 μm. p-values are indicated by asterisks with $^{*}$p<0.05, $^{**}$p<0.01, $^{***}$p<0.001.
(TIF)

**S4 Fig. The late germ cell marker Boule is expressed in testes of *zpg*::GFP GR and phospho mutant flies, indicating normal germ cell differentiation.** (A-D) Testes stained for the meiotic germ cell marker Boule (red) and nuclei (blue). In testes of wildtype flies (A) or from *zpg* mutants rescued with *zpg*::GFP GR (B), which show a complete rescue of the *zpg* null phenotype, Boule signal is found in late germ cell cysts as well as in long and parallel spermatid bundles. In *zpg* null mutants (C), meiotic stages are not reached hence no Boule signal is detected. Testes of both phosphorylation site mutants (*zpg* Y352F in D, *zpg* Y352F/S356A in E) show strong Boule signal and the parallel organization of spermatid bundles. Hubs are marked by asterisks. (F) Testes stained for nuclear marker (white), arrows indicate elongated spermatid bundles. Scale bars represent 50 μm.
(TIF)

**S5 Fig. Strong defects and absence of membrane localization is observed in testes of GFP-tagged *zpg* cysteine mutants.** (A-L) Immunostaining for germ and somatic cells in wild type (A-C), *zpg* null mutant (D-F), *zpg* null mutant rescued with *zpg*::GFP C6S (G-I) and *zpg* null mutant rescued with *zpg*::GFP C26S (J-L). Both C6S::GFP and C26S::GFP rescue exhibit a phenotype that is indistinguishable from the *zpg* null mutant with a decreased number of germ cells (Vasa+) and late somatic cells (Eya+) as well as an increased number of early-mid somatic cells (Zfh-1+ and Tj+). The immunostainings, as well as the quantification of (M) germline stem cells (GSCs), (N) Zfh-1-positive cells, (O) Tj-positive cells and (P) Eya-positive cells reveals a strong, nearly null-mutant like phenotypes in *zpg*::GFP C6S and *zpg*::GFP C26S rescued testes, indicating a strong loss of function. This is consistent with the absence of the mutated Zpg::GFP C6S and Zpg::GFP C26S proteins from the plasma membrane (S-U). (Q-T) Colocalization of endogenous, unmutated Zpg and GFP-tagged, mutated Zpg in flies with one copy of endogenous Zpg. *zpg* GFP::GR and *zpg* heterozygous controls are depicted in B-B' and C-C', respectively. In testes expressing the *zpg*::GFP C6S (S-S') and *zpg*::GFP C26S (T-T") mutant constructs, the GFP-tagged mutated Zpg accumulates intracellularly, while endogenous Zpg mainly localizes to the plasma membrane. This results in low Pearson colocalization coefficients (U) upon quantification of the colocalization of endogenous and mutated Zpg testes of flies with the *zpg*::GFP C6S and *zpg*::GFP C26S mutant transgenes compared to the *zpg*::GFP GR control.
(TIF)

**S6 Fig. Analysis of Zpg localization and somatic cell development in *zpg* D50 point mutants.** (A-E) The subcellular localization of mutant Zpg proteins is revealed by staining for GFP (green) as all rescue transgenes contain a GFP tag at the C-terminus of Zpg, Fas3 (red) is

used to mark the hub and nuclei are stained with DAPI (blue). GFP single channel is depicted in in white (A'-E'). *zpg*::GFP GR rescue control (A-A') shows GFP localization at the membrane. *Zpg* null mutants (B-B') do not express GFP. In testes of *zpg* mutants rescued with the *zpg* D50A (C-C') or *zpg* D50R (D-D') rescue constructs, which show mild germ cell differentiation defects, the GFP signal is strongly enriched at the plasma membrane, indicating normal localization of the mutant GJ proteins. The low number of germ cells in testes of *zpg* mutants rescued with the *zpg* D50K mutant rescue construct (E-E') makes it hard to determine the localization of the GFP-tagged mutant Zpg protein. Therefore, localization of this mutant was analyzed in a heterozygous background, containing one copy of the endogenous *zpg*, this data is shown in Fig 5. (F-T) Analysis of germ cell and somatic cell markers in the N-terminal mutant rescue constructs. Staining for Vasa (mitotic germ cells, left and right panel; green) and the somatic markers Zfh-1 (early soma, left panel; grey), Tj (early-mid soma, middle panel; magenta) and Eya (late soma, right panel; magenta). Hubs are marked with Fas3 in grey, nuclei are labelled in blue. Wild type control depicted in F-H, *zpg* null mutant in I-K. The number of early somatic cells (Zfh-1+, Tj+) in the testes of *zpg* mutants rescued with either *zpg* D50A (L, M), *zpg* D50R (O, P) or *zpg* D50K rescue constructs (R, S). All analyzed mutants have a lower number of Eya+ cells than wt (right panel). Associated quantifications are shown in Fig 5. Hubs are marked with dashes. Scale bars represent 50 μm.
(TIF)

**S7 Fig. Analysis of Zpg localization and somatic cell development in *zpg* D59 point mutants.** (A-E) The subcellular localization of mutant Zpg proteins is revealed by staining for GFP (green) as all rescue transgenes contain a GFP tag at the C-terminus of Zpg, Fas3 (red) is used to mark the hub and nuclei are stained with DAPI (blue). GFP single channel is depicted in in white (A'-E'). *zpg*::GFP GR rescue control (A-A') shows GFP localization at the membrane. *zpg* null mutants (B-B') do not express GFP. The low number of germ cells in testes of *zpg* mutants rescued with the *zpg* D59A mutant rescue construct (E-E') makes it hard to determine the localization of the GFP-tagged mutant Zpg protein. In testes of *zpg* null mutants rescued with the *zpg* D59N (D-D') or the *zpg* D59H (E-E') rescue transgenes, the GFP signal is mainly concentrated at the membrane. Therefore, localization of these mutants was analyzed in a heterozygous background, containing one copy of the endogenous *zpg*, this data is shown in Fig 6. (F-T) Analysis of germ cell and somatic cell markers in the N-terminal mutants. Staining for Vasa (mitotic germ cells, left and right panel; green) and the somatic markers Zfh-1 (early soma, left panel; grey), Tj (early-mid soma, middle panel; magenta) and Eya (late soma, right panel; magenta). Hubs are marked with Fas3 in grey, nuclei are labelled in blue. Wildtype control is depicted in F-H, *zpg* null mutant in I-K. In testes of *zpg* null mutants rescued with the D59A rescue construct a strong somatic cell differentiation defect was seen with elevated cell counts for the early somatic markers Zfh-1 (L) and Tj (M), whereas these cell counts appear wildtype for zpg nulls mutants rescues with either the *zpg* D59N (O, P) or *zpg* D59H (R, S) rescue construct. All analyzed mutants have a lower number of Eya+ cells compared to wt (right panel). Associated quantifications are shown in Fig 6. Hubs are marked with dashes. Scale bars represent 50 μm.
(TIF)

**S8 Fig. Analysis of Zpg localization and somatic cell development in N-terminal mutants.** (A-E) The subcellular localization of mutant Zpg proteins is revealed by staining for GFP (green) as all rescue transgenes contain a GFP tag at the C-terminus of Zpg, Fas3 (red) is used to mark the hub and nuclei are stained with DAPI (blue). GFP single channel is depicted in in white (A'-E'). *zpg*::GFP GR rescue control (A-A') shows GFP localization at the membrane. *zpg* null mutants (B-B') do not express GFP. The low number of germ cells in testes of *zpg*

mutants rescued with either the *zpg* D21A (C-C'), *zpg* D21N (D-D'), or *zpg* delta2-5 rescue constructs (E-E') makes it hard to determine the localization of the GFP-tagged mutant Zpg protein. Therefore, localization of these mutants was analysed in a heterozygous background, containing one copy of the endogenous *zpg*, this data is shown in Fig 7. (F-T) Analysis of germ cell and somatic cell markers in the N-terminal mutants. Staining for Vasa (mitotic germ cells, left and right panel; green) and the somatic markers Zfh-1 (early soma, left panel; grey), Tj (early-mid soma, middle panel; magenta) and Eya (late soma, right panel; magenta). Hubs are marked with Fas3 in grey, nuclei are labelled in blue. Wild type control depicted in F-H, *zpg* null mutant in I-K. Testes of *zpg* mutants rescues with the *zpg* D21A (L-N), *zpg* D21N (O-Q) or *zpg* delta2-5 (R-T) mutants show a lower number of Vasa-positive germ cells and Eya-positive late somatic cells, while at the same time the number of early somatic cells (Zfh-1- and Tj-positive) is increased. This phenotype, seen in all three N-terminal mutants is indistinguishable from that of the *zpg* null. Associated quantification are shown in Fig 7. Hubs are marked with dashes. Scale bars represent 50 μm.
(TIF)

**S9 Fig. Colocalization of endogenous and mutant Zpg proteins.** Representative scatter plots generated with the JaCoP plugin in FIJI, one shown per genotype. The signal intensity for endogenous Zpg is plotted against the signal intensity of the GFP-tagged mutated Zpg in flies that have one copy of endogenous and one copy of mutated protein. A linear relationship between endogenous and GFP-tagged signal intensity, like in *zpg*::GFP GR flies (A) indicates strong colocalization of mutated and endogenous Zpg at the membrane. For *zpg* D21A (B) and *zpg* D21N (C) a weak colocalization is detected. Strong colocalization, similar to the control in A, is found in heterozygotes of *zpg* delta2-5 (D), *zpg* D50A (E), *zpg* D50R (F), *zpg* D50K (G), *zpg* D59A (H), *zpg* D59H (I) and *zpg* D59N (J), whereas colocalization is minimal for *zpg* C6S::GFP (K) and *zpg* C26S::GFP (L).
(TIF)

## Author Contributions

**Conceptualization:** Yanina-Yasmin Pesch, Guy Tanentzapf.

**Data curation:** Yanina-Yasmin Pesch.

**Formal analysis:** Yanina-Yasmin Pesch, Vivien Dang, Fayeza Islam, Priya Kaur, Ciaran R. McFarlane.

**Funding acquisition:** Filip Van Petegem, Guy Tanentzapf.

**Investigation:** Yanina-Yasmin Pesch, Vivien Dang, Fayeza Islam, Priya Kaur, Christopher M. Smendziuk, Ciaran R. McFarlane, Pierre-Yves Musso.

**Methodology:** Michael John Fairchild, Darius Camp, Christopher M. Smendziuk, Anat Messenberg.

**Project administration:** Guy Tanentzapf.

**Resources:** Michael John Fairchild, Darius Camp, Christopher M. Smendziuk, Anat Messenberg.

**Software:** Rosalyn Carr.

**Supervision:** Filip Van Petegem, Guy Tanentzapf.

**Validation:** Yanina-Yasmin Pesch.

**Visualization:** Rosalyn Carr, Ciaran R. McFarlane.

**Writing – original draft:** Yanina-Yasmin Pesch, Guy Tanentzapf.

**Writing – review & editing:** Yanina-Yasmin Pesch, Guy Tanentzapf.

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
