## [Decision Letter · Decision Letter 0]

9 Jul 2022

Dear Guy,

Thank you very much for submitting your Research Article entitled 'Gap Junctions Mediate Discrete Regulatory Steps During Fly Spermatogenesis' to PLOS Genetics.

We are pleased to say that all 3 reviewers sent back positive reviews on your manuscript. However, they made some valuable and constructive comments, that you should consider in a revised version of your manuscript. Please send us a rebuttal letter addressing the reviewers’ comments and a revised version of your manuscript. Then, we will be happy to accept your manuscript for publication.

We therefore ask you to modify the manuscript according to the review recommendations. Your revisions should address the specific points made by each reviewer.

[LINK]

Yours sincerely,

Jean-René Huynh

Associate Editor

PLOS Genetics

Gregory P. Copenhaver

Editor-in-Chief

PLOS Genetics

Reviewer's Responses to Questions

**Comments to the Authors:**

Reviewer #1: Pesch et al. reports their study on the role of a gap junction protein (Zpg) in the Drosophila spermatogenesis. The role of gap junction-mediated signaling is well known to regulate soma-germline communication (and cell-cell communication, more broadly) but the molecular details remained underexplored. This manuscript presents a thorough study, which generated many point mutations to a gap junction protein (Zpg). They used available structure information to design point mutations in Zpg protein and examined their ability to rescue zpg mutant (and examined mutant phenotypes when the rescue transgene is not fully functional). The study indicates that different mutations affect different stages of germ cell development, providing novel and important insights into how soma-germline interaction carve the process of germ cell differentiation. Specifically, they generate mutants for C-terminal deletion, phosphorylation mutants, Cysteine mutants that are proposed to interfere with GJ formation or cargo selectivity. The results are quite informative and contribute significantly to the field.

The study also has an important implication as to why mammalian Sertoli cells, which encapsulate all stages of germ cells, can regulate different steps of germ cell differentiation. This (why single Sertoli cell can regulate spermatogonia and spermatocytes and spermatid at the same time) has been a question many people vaguely asked without any idea. Discussing their finding in this context would be nice.

Despite these very positive aspects of the paper (which exceed the standard by any means to be published in PLOS Genetics), the writing is confusing here and there, likely impeding the appreciation of the work by many readers. My comments are all minor, but I think they will improve the quality of the writing (I may be confused as well by the writing style, so I do not ask them to change the text the way I suggested. But hope they see why a reader can get confused, and correct the writing to be more understandable to broader readers).

Minor issues

-Intro 2nd page, 2nd paragraph is extremely confusing. It starts out by calling Innexins and connexins as ‘gap junction’ proteins (where they state that GJ is the connection between hemichannels between two adjacent cells). Then go on to say that innexin’s ORTHOLOG are pannexins, which only forms hemi-channel. The definition of ortholog is they are functionally equivalent. Thus, it reads as if the authors are committing to saying that innexins function as hemi-channel---but the rest of the intro is written as if innexins and connexins are functional equivalents (=orthologs), and much later in the results, they say it is not well known which is the mode of function for innexins. Combination of possible misuse of the term and not-so-straightforward introduction of information throughout the manuscript makes the manuscript quite confusing.

-Fig1B: should have information that inx4 is zpg (not just in the legend/text).

-Fig2D: I believe that Zpg-GFP colocalizes with endogenous Zpg, but this figure does not quite prove that point. In this genotype, there are Zpg-GFP transgene and wild type (endogenous) Zpg gene. When anti-Zpg is used, it will recognize Zpg-GFP as well as endogenous Zpg protein. And this Zpg-GFP molecule that is also recognized by the antibody will surely co-localize with GFP. Unless endogenous Zpg localizes very different place (such as nucleus, nuclear envelope, ER), one cannot prove ‘colocalization’ by GFP and antibody staining. I am not saying they are wrong, but the point that Zpg-GFP colocalizing with endogenous Zpg can be made simply by comparing Zpg-GFP localization and anti-Zpg staining of the wt. I am not requesting any experiment, but the authors might want to modify the text. Only after reading to the end of Fig4, it is explained that C-ter GFP tag of Zpg protein interferes the recognition by anti-Zpg antibody. This info should actually be provided earlier here.

-Fig3E vs. 3E”: I don’t know about the journal style, but generally speaking, the images shouldn’t be given the same letter (E vs. E”) if they are different images.

-Y352F S356A: I don’t like that the double mutant called ‘phospho-dead’---because this term usually refers to a single site (Y352F is also ‘phospho dead’ of Y352F). Also unless phosphorylation status is checked by mass-spec etc., the double mutant Y352F S356A could have some other phosphorylation. One does not need to name a mutant vaguely (which can be misleading), but just calling ‘Y352F, S356A’ is much more accurate.

-Based on the images presented in Fig3C and D, localization of Y352F and Y352F S356A seems to be altered (as opposed to the statement in the manuscript). The mutants’ signal on the membrane seems to be less contiguous (patches and foci more prominent than wt). I agree with their statement about that the spermatogenesis appears normal (at least up until meiosis).

-Fig4: they generate mutants that are supposed to be defective in forming GJ (meaning, the hemichannels cannot engage in interaction with the hemichannels on the other cell). There are a few confusing issues/problems in this figure. First, if panel A-D is correct, Zpg mutants are not expressed at all (or very weakly expressed). This suggests that, these mutations might just interfere with the stability of the protein (before forming channels), and perhaps their conclusion that Zpg functions by forming channels may not be as strongly supported as stated. The text should be modified accordingly. The lack of Zpg Cys mutant expression is somewhat inconsistent with Zpg-Cys-GFP results shown in Fig4E (clearly the same mutant, being only different by the tag).

-Fig5-7: it is hard to follow when they talk about the data from these 3 figures, going back and forth.

-Fig5: panel L and M cannot be the same category of the phenotype (in L, the testis is filled with pre-meiotic germ cells, whereas the testis in M contains elongating spermatids). I imagine these images reflect sterile vs. fertile testes of D50A mutants, but it should be explained more carefully.

Reviewer #2: The authors have performed an elegant structure function analysis of the gap junction protein zero population growth (zpg). Prior work from the Tanentzapf lab and other labs has shown that Zpg is expressed in the germline, that zpg mutants have rudimentary gonads, and that Zpg in germ cells interacts with Innexin2 (Inx2) in somatic support cells. The Tanentzapf lab has previously shown that zpg-/- phenotypes could be rescued fully by an exogenous transgene containing the zpg genomic locus inserted at an attB site. This transgene is the workhorse of the current study. The authors make site-directed mutations to the C-terminal domain, the N-terminal domain, phosphorylated Ser and Tyr resides, Cys residues in hemichannel in the extracellular domain, and Asp residues in the channel pore. They then analyzed the ability of these transgenes to rescue germline and somatic defects of the zpg-/- null testes. They find that the C-terminus is required for Zpg to localize to the plasma membrane; that the conserved phosphoresidues are not required for Zpf function; that extracellular Cys residues are involved in membrane localization or protein stability; that Asp mutations produced distinct phenotypes. When the mutation caused a strong phenotype, the effects on GSC maintenance, germline differentiation and somatic differentiation were similar to each other and were similar to the zpf null phenotype. However, weaker "allele" produced intermediate phenotype. Interestingly, D50A and D59A caused defects in spermatid individualization. Overall, the work was of high quality, the data were compelling, all controls were included, the quantification was robust and the work revealed new insights into gap junction function.

I have only a couple of comments about the writing. (And on this note, would the authors please provide line numbers and page numbers even if this is not required by the journal because these two things really help the reviewer when writing the critique.)

1. On p 13 of the PLoS Genet assembled pdf under "Methodology for assessing rescue ability of rescue constructs", this is not a sentence and should be revised: "This reduction meant in zpg mutants rescued with one copy of zpg::GFP GR express Zpg at levels that similar to those seen in heterozygous zpg mutants

(zpg2533/+) where Zpg antibody staining intensity was 34.4% lower compared to that seen in

homozygous wildtype testes (Fig. S2)".

2. In the first paragraph of the discussion, the authors write: " Spermatogenesis is particularly suitable for this type of structure function, as encapsulation of the germ cells by somatic cells completely isolates the soma from the environment (Schulz et al., 2002; Fairchild et al., 2015). This means that the soma is fully reliant for its survival on gap junction mediated transport of external cues required for cell differentiation and/or nutrients and

metabolites." In my understanding, one face of the somatic cells are exposed to the testis lumen and so in fact are not completed isolated and are not completely dependent on germ cells for nutrients. Can the authors please provide additional support for their statement?

3. Fig. 5 - the circled "tumor" is so small that it is not legible.

Reviewer #3: In this study, Pesch and colleagues present a structure-function analysis of the germ cell-specific Innexin, Zpg. They use an elegant genomic rescue method to assay an impressive set of mutations in various domains of the protein for function, exploiting the fact that germ cell differentiation is easily observable and provides an excellent readout for gap junction function. Using this approach, they identify several critical domains and residues for Zpg function, as well as novel, late phenotypes that were previously masked by the earlier effects of null mutations.

The authors relied extensively on a model of Zpg structure, which was obtained by homology with the only Innexin for which a cryo-EM derived structure exists, from the nematode C Elegans. One issue with this manuscript is that the model is not described in sufficient detail, and the assumptions made in the modelling are not made clear. This makes it hard to judge the extent of confidence in the homology model and how much of the model is speculative. Could the authors provide confidence values for the localization of the specific residues that are predicted to be within the pore domain? Since this model-derived structure is still hypothetical, the authors should use more cautious language in describing the residues’ localization or structural function. A similar caution would be most welcome when describing the tyrosine residues as “conserved phosphorylation sites”. Given that there is no evidence presented that these residues are actually phosphorylated and that mutating them does not result in any germ cell or fertility phenotype, the authors should refer to these as “conserved tyrosine residues”.

Several mutants show impaired protein localization, which the authors refer to as “cytoplasmic”. Is this indeed cytoplasmic or is the protein within membranes in internal vesicles? None of the mutants should impact on the signal peptide, suggesting that all proteins should be incorporated into the ER during translation. Could the authors address this? Similarly, in the case of the C-terminal deletion (and some others), there is little overlap between the localization of the mutant form of the protein and the endogenous, when both are present. However, the model suggests that Zpg should multimerize, and the residues that are mutated should not impact on this interaction domain which, to this reviewer’s understanding, is within the transmembrane domain. Could the authors speculate and/or exploit their model to predict how the various mutants should impact Zpg interactions with itself? Alternatively, is the defect in trafficking of the mutant proteins, and if so, how does that impact our understanding of the assembly of these multimeric complexes?

Some of the mutants (Cysteine mutants and C-terminal deletion) partially rescue GSC numbers but no other phenotypes. Can the authors speculate why this would be? Is the localization of these mutants similar to endogenous localization in GSCs which does not appear to be at the membrane/interface with CySCs? This could be assayed by comparing the colocalization specifically in GSCs vs other germ cells. The text mentions that “the zpg cysteine mutant behave indistinguishably from null alleles of zpg”, however the partial GSC number rescues would indicate otherwise.

Additionally, the images for the Cysteine mutants consistently show an enlarged hub (as do occasional others). Although this is a known phenotype of germ cell loss (PMID: 8756289), this does not appear to be consistently the case in the zpg mutant alone. Is this simply a reflection of the choice of images, or is there a different effect on hub cell size between the null and Cysteine mutants?

Minor issues:

“Vasa is a marker that labels mitotic germ cells, meaning the early stages of spermatogenesis, from germline stem cells (GSCs) to spermatocytes” – spermatocytes are not mitotic.

The authors should show an image of DAPI-positive sperm bundles and explain how counts were done, maybe by highlighting what constitutes a sperm bundle on the image.

More detail is required for the fertility assays. The methods indicate that the number of offspring were counted but the graph only shows “fertile” or “subfertile”. How were these categories assessed?

Cysteine mutants – the Zpg channel is not visible in Figs. 4B-D.

“D to A mutations constitute the most significant functional change within our mutagenesis approach” – this is based on the prediction, however the phenotypes do not match this prediction, at least for D50, and arguably also D21. Please amend the text to make clear that this is a prediction, not an observation.

D59N appears more cytoplasmically-localized, although some is still visible at the membrane. Is this correct or simply the image chosen? The text indicates that D59N localized normally.

Typo p. 26 “ the soma is fully reliant for its survival on gap junction- mediated transport of external cues”. Did the authors mean the germline?

**Have all data underlying the figures and results presented in the manuscript been provided?**

Reviewer #1: Yes

Reviewer #2: Yes

Reviewer #3: Yes

PLOS authors have the option to publish the peer review history of their article (what does this mean?). If published, this will include your full peer review and any attached files.

Reviewer #1: **Yes: **Yukiko Yamashita

Reviewer #2: No

Reviewer #3: No

---

## [Editor Report · Decision Letter 1]

7 Sep 2022

Dear Guy,

We are pleased to inform you that your manuscript entitled "Gap Junctions Mediate Discrete Regulatory Steps During Fly Spermatogenesis" has been editorially accepted for publication in PLOS Genetics. Congratulations!

Yours sincerely,

Jean-René Huynh

Academic Editor

PLOS Genetics

Gregory P. Copenhaver

Editor-in-Chief

PLOS Genetics

Comments from the reviewers (if applicable):

**Data Deposition**

http://datadryad.org/submit?journalID=pgenetics&manu=PGENETICS-D-22-00662R1

**Press Queries**

---

## [Editor Report · Acceptance letter]

26 Sep 2022

PGENETICS-D-22-00662R1 

Gap Junctions Mediate Discrete Regulatory Steps During Fly Spermatogenesis 

Dear Dr Tanentzapf, 

We are pleased to inform you that your manuscript entitled "Gap Junctions Mediate Discrete Regulatory Steps During Fly Spermatogenesis" has been formally accepted for publication in PLOS Genetics! Your manuscript is now with our production department and you will be notified of the publication date in due course.

With kind regards,

Agnes Pap

PLOS Genetics

On behalf of:
